# Interdomain-linkers control conformational transitions in the SLC23 elevator transporter UraA

Benedikt T. Kuhn [1,2], Jonathan Zöller [3], Iwan Zimmermann [4], Tim Gemeinhardt[1], Dogukan H. Özkul [1], Julian D. Langer[3,5], Markus A. Seeger [4] & Eric R. Geertsma [1,2] ✉

Uptake of nucleobases and ascorbate is an essential process in all living organisms mediated by SLC23 transport proteins. These transmembrane carriers operate via the elevator alternating-access mechanism, and are composed of two rigid domains whose relative motion drives transport. The lack of large conformational changes within these domains suggests that the interdomain-linkers act as flexible tethers. Here, we show that interdomain-linkers are not mere tethers, but have a key regulatory role in dictating the conformational space of the transporter and defining the rotation axis of the mobile transport domain. By resolving a wide inward-open conformation of the SLC23 elevator transporter UraA and combining biochemical studies using a synthetic nanobody as conformational probe with hydrogen-deuterium exchange mass spectrometry, we demonstrate that interdomain-linkers control the function of transport proteins by influencing substrate affinity and transport rate. These findings open the possibility to allosterically modulate the activity of elevator proteins by targeting their linkers.

Biological membranes efficiently divide space into two compartments. The transport proteins residing in these membranes control the essential exchange of solutes and thereby directly affect the internal milieu of cells and organelles. With more than 450 members distributed in 65 subfamilies, solute carriers (SLCs) form the largest group of membrane transport proteins[1,2]. SLCs are secondary transporters that rely on electrochemical gradients across membranes to drive solute transport. One of the three main transport modes discerned for SLCs is the so-called elevator alternating-access mechanism[3]. A plethora of protein structures[4] established that the unifying design of the structurally diverse elevator-type transporters involves a mobile transport domain and a static scaffold domain. Of these, the transport domain accommodates the substrate binding site and traverses the membrane. In contrast, the scaffold domain remains stably embedded in the membrane and provides the homo- or hetero-oligomerization interface. During transport both domains remain essentially rigid and only undergo a relative reorientation. Despite this structural rationale for elevator-type transport based on protein structures of transporters in different conformations, the basic design principles underlying the individual transporters' conformational space and temporal dynamics are not clear.

The solute carrier families 4, 23, and 26 contribute to essential physiological processes, such as ion homeostasis and vitamin transport, illustrated by their causative roles in certain pathologies[5–7]. The structure of the *E. coli* uracil permease UraA defined a new, unique architecture based on a 7-transmembrane segment inverted-repeat (7TMIR) fold[8,9] for the SLC23 family that is shared by the transmembrane domains of the SLC4 and SLC26 family[10,11]. Based on a detailed analysis of the internal pseudo-symmetry of the inverted repeats[9,12], structures of individual SLC23, SLC4, and SLC26 proteins in different

[1]Institute of Biochemistry, Biocenter, Goethe University Frankfurt, Frankfurt am Main, Germany. [2]Max Planck Institute of Molecular Cell Biology and Genetics, Dresden, Germany. [3]Proteomics, Max Planck Institute of Biophysics, Frankfurt am Main, Germany. [4]Institute of Medical Microbiology, University of Zurich, Zurich, Switzerland. [5]Proteomics, Max Planck Institute for Brain Research, Frankfurt am Main, Germany. ✉e-mail: geertsma@mpi-cbg.de

conformations[8,10,11,13–25], and recent structures of human SLC4A2 in the inward-open, outward-open, and occluded conformation[26], it has become clear that these families operate according to the elevator alternating-access mechanism.

Members of the SLC23 family are responsible for the uptake of vitamin C in mammals and of nucleobases in all kingdoms of life[5]. Like all proteins with this fold, SLC23 proteins form dimers and dimerization is relevant for transport[13,14,27]. UraA is the only SLC23 protein for which structures in multiple conformations are known[8,13]. Superimposition of the inward-open and occluded conformation of UraA revealed that transport relies on a rigid body movement combining translation and rotation of the transport domain relative to the scaffold domain, in line with the elevator-type transport mode[13]. However, the structural motifs coordinating this transition have remained unclear[9]. The transport and scaffold domains in UraA are physically connected by interdomain-linkers that take the shape of short, partially alpha-helical tethers. This motif is common in elevator transporters: it is not only observed in the SLC4, SLC23, and SLC26 families, but also in SLC9, SLC10, SLC13, 2HCT, and AbgT transporters[28–32]. Despite the extensive structural characterization of elevator transporters in the last years[4], little attention has been paid to the analysis of interdomain-linkers and their functional relevance in elevator transport.

Here, by combining structural and biochemical studies with hydrogen-deuterium exchange mass spectrometry (HDX-MS), we assess the relevance of the interdomain-linkers connecting the rigid transport and scaffold domain in elevator-type transporters. Using UraA as a model system, we identify common design elements in the interdomain-linkers. These do not only define the rotation axis of the mobile transport domain but also coordinate the spatiotemporal dynamics of the conformational transitions of the transporter, thereby strongly impacting functional parameters such as transport rate and substrate affinity. As a consequence, this qualifies interdomain-linkers as a new platform for allosteric modulation of elevator-type transport proteins.

## Results

### Molecular hinges in SLC23 transporters

Thus far, UraA is the only member of the SLC23 family for which more than one conformation has been structurally characterized. Comparison of the inward-open (UraA$_{IO}$) and occluded (UraA$_{OCC}$) structure of UraA highlighted that this conformational change essentially consists of relative reorientations of two rigid bodies, the transport and the scaffold domain[8,13] (Fig. 1A). Following this logic, the actual deformations in the peptide backbone that are required for transport should be limited to local rearrangements in the tethers connecting both domains.

As for most elevator proteins, three interdomain-linkers can be discerned in SLC23 transporters[3,13,14,33] (Fig. 1B). Of these, the long loop TM7-8, which connects the inverted repeats, appears an unstructured

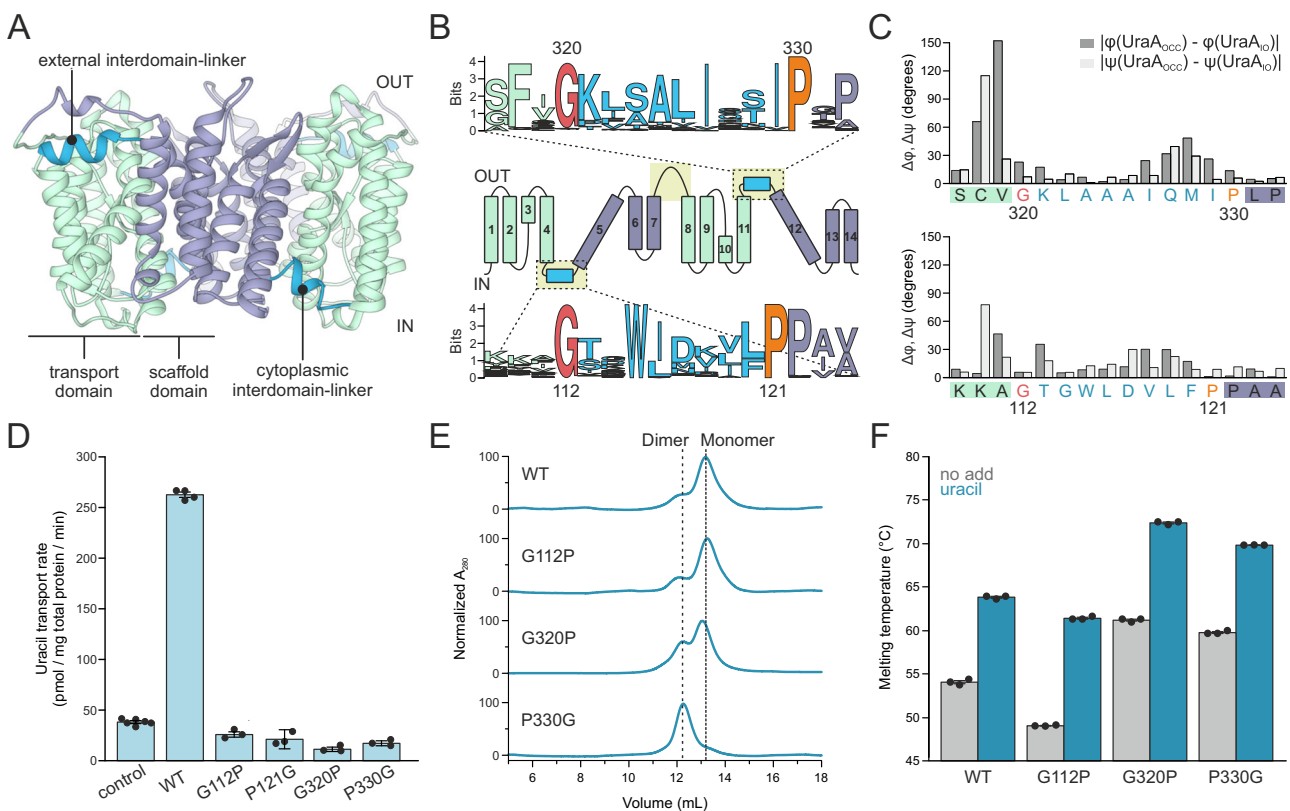

**Fig. 1 | Functional analysis of inter-domain linker mutants in UraA. A** Dimeric UraA structure (PDB:5XLS) with scaffold and transport domain in green or purple, respectively, and the interdomain-linkers in blue. **B** Topology plot of UraA (center) with color code as in (A). Upper and lower panels depict the sequence conservation of the extracellular and cytoplasmic interdomain-linkers in the UraA-subfamily, respectively, shown as sequence logo. **C** Absolute differences in dihedral angles between interdomain-linker residues in UraA$_{IO}$ (PDB:3QE7) and UraA$_{OCC}$ (PDB:5XLS). Δφ and Δψ are colored dark and light gray, respectively, and glycine and proline residues flanking the spacer helices are highlighted in red and orange, respectively. **D** In vivo uptake rates of [³H]-uracil by wild type UraA and interdomain-linker mutants upon expression in *E. coli* BW25113(*ΔuraA*) with technical replicates shown as scatter plot and derived mean values ± SEM as bars (control: *n* = 6, WT: *n* = 4, G112P, P121G, G320P, and P330G: *n* = 3). A UraA variant with three alanine substitutions in the substrate binding site (E241A, H245A, and E290A) served as negative control. **E** Representative size-exclusion chromatograms of decylmaltoside-solubilized UraA variants in the absence of substrate. **F** Melting temperature of interdomain-linker mutants as determined by differential scanning fluorimetry in absence and presence of 1 mM uracil with technical replicates shown as scatter plot and derived mean values ± SEM as bars (*n* = 3 for all samples).

and flexible tether. In contrast, the two short interdomain-linkers TM4-5 and TM11-12 are mostly composed of a rigid alpha-helix that is also present in other elevator transport families (Supplementary Fig. 1). During transport, the short loops flanking these alpha-helical regions are thus expected to bend, thereby acting as molecular hinges that potentially define the conformational trajectory of the transport cycle.

In UraA, the complete TM4-5 and TM11-12 linkers measure approximately ten residues that accommodate the short amphipathic alpha-helix of two to three helical turns, termed 'spacer helix', flanked by one or two amino acids. A comparison of the dihedral angles of these residues in $UraA_{IO}$ and $UraA_{OCC}$ indicates that this region undergoes local conformational changes in the peptide backbone (Fig. 1C). These changes are most pronounced on the side of the transport domain.

Near the scaffold domain both spacer helices are flanked by a proline that is strongly conserved in the SLC23 (Fig. 1B and Supplementary Fig. 2), but also in the external linker of the structurally-related SLC4 and SLC26 families (Supplementary Fig. 3)[8,10,11,13–24]. On the opposite side near the transport domain the spacer helix is flanked by a conserved glycine in the UraA-subfamily. The unique conformational properties of glycine (flexible) and proline (restricted) coincide well with the degree of local conformational changes observed in the peptide backbone on either side of the spacer helices in UraA.

## Mutations in interdomain-linkers affect the function

To study the role of the glycine and proline residues flanking the spacer helices in more detail, we created the UraA variants G112P and P121G in the cytoplasmic interdomain-linker, and G320P and P330G in the external linker. For all mutants, uracil transport in whole cells was severely compromised (Fig. 1D), but only for UraA(P121G) this could be accounted for by strongly reduced expression levels (Supplementary Fig. 4). To determine the basis for the inactivity of the other mutants, we purified wild type UraA and the respective variants and assessed the overall folding state and ability to bind uracil. Previous studies identified a loss of UraA transport activity due to monomerization[13] or hampered substrate binding[8]. Decylmaltoside-solubilized wild-type UraA eluted in two peaks previously assigned to UraA monomers and dimers[13]. For both wild type UraA and UraA(G112P), the monomeric species was the most abundant (Fig. 1E and Supplementary Fig. 4B, C, and Supplementary Table 1). In contrast, the dimer content was increased in UraA(G320P) and dimers were the dominant species in UraA(P330G). The presence of uracil did not alter the relative distribution of these populations (Supplementary Figs. 4B and 4D). Since constitutive UraA dimers have high transport activities[13], it is not obvious why the UraA(G320P) and UraA(P330G) mutants are inactive.

We further qualitatively assessed uracil binding by differential scanning fluorimetry (Fig. 1F and Supplementary Fig. 5). Wild-type UraA showed a melting temperature of $54.2 \pm 0.2\,°C$ that was increased to $64.1 \pm 0.1\,°C$ in the presence of 1 mM uracil. A similar degree of ligand-induced thermostabilization was observed for the G112P, G320P, and P330G mutants, indicating that these variants are still capable of binding substrate. Furthermore, these data are consistent with previous observations that the oligomeric state itself does not affect uracil-binding properties[13]. Noteworthy, while the thermal stability of UraA(G112P) was reduced compared to the wild-type protein, we already observed a strong increase in melting temperature for both apo-UraA(G320P) and -UraA(P330G) that upon addition of uracil exceeded $70\,°C$. The substitutions G320P and P330G thus led to a significant thermostabilization of UraA along with an increased dimer stability. Remarkably, for both variants this was achieved by only a single amino acid substitution.

## Validating a synthetic nanobody as conformational probe

As the loss of function in UraA(G320P) and UraA(P330G) cannot be attributed to monomerization (Fig. 1E) or impaired substrate binding (Fig. 1F), we focused on uncovering potential anomalies in the conformational dynamics of these UraA mutants. We identified sybody Sy45 from our synthetic single-domain antibody library[34] as a potential tool to explore the conformational space of UraA. Sy45 was selected against wild-type UraA and expected to be conformational-selective based on its pronounced thermostabilization of unliganded UraA by $16.7\,°C$ (Supplementary Fig. 6).

We studied the conformational specificity of Sy45 by co-crystallization with wild-type UraA and UraA(G320P). The stabilizing effect of Sy45 was sufficient to allow the structural characterization of an SLC23 protein in the absence of added ligand. Sy45 co-crystallized with wild-type UraA and UraA(G320P) under similar conditions, but crystals of the latter were optimized more efficiently resulting in a structure refined to 3.5 Å (Fig. 2; Supplementary Figs. 7 and 8; and Supplementary Table 2). In the UraA(G320P)-Sy45 structure, Sy45 binds a UraA protomer from the cytoplasmic side. In this conformation, UraA displays a large cleft between the scaffold- and transport domain open to the cytoplasm that allows access of the bulk solvent to the substrate binding site (Fig. 2B). Thus, Sy45 stabilizes an inward-facing conformation of UraA. All three complementarity-determining regions of Sy45 contribute to binding (Fig. 2C). Together, they cover an extensive epitope of 1095 Å$^2$ that includes TM8 and TM9 of the transport domain[35], TM5 and TM12 of the scaffold domain, and the cytoplasmic interdomain-linker. The position of Sy45, wedged in between the transport and scaffold domain, prevents the relative reorientation of the transport domain and thereby conformationally arrests UraA but still provides ample space for uracil to reach the substrate binding site (Fig. 2B). This is also reflected by a second crystal structure of UraA(G320P)-Sy45 in presence of uracil refined to 3.7 Å showing the transporter in the same inward-facing conformation (Supplementary Fig. 8).

We compared our new UraA(G320P)-Sy45 structure with the existing $UraA_{IO}$[8] and $UraA_{OCC}$[13] structure, respectively. The transport domain of UraA(G320P)-Sy45 superimposes well with the respective domain of both $UraA_{IO}$ and $UraA_{OCC}$, resulting in a root mean square deviation (RMSD) of 0.6 and 0.7 Å, respectively (Fig. 3A and Supplementary Fig. 9A). Superimposition of the scaffold domains only yielded a comparably low RMSD of 1.9 Å for $UraA_{OCC}$. In contrast, alignment with the scaffold domain of $UraA_{IO}$ revealed a major deviation in the position of TM6 and TM7s (Supplementary Fig. 9B). Of note, this region of $UraA_{IO}$ also showed poor alignment with $UraA_{OCC}$ and the inward-open structure of UapA[13,14]. With an RMSD of 4.1 Å over 358 residues, our structure resembles that of inward-facing UapA (Supplementary Figs. 9C and 10)[14]. It displays a wider opening and consequently a larger cytoplasmic cavity (4780 Å$^3$) than $UraA_{IO}$ (3300 Å$^3$)[36,37], hence we term this conformation wide inward-open ($UraA_{WIO}$) (Supplementary Figs. 10 and 11).

## Hinges in interdomain-linkers underlie conformational transition

The high degree of structural similarity between the transport and scaffold domains of $UraA_{WIO}$ and $UraA_{OCC}$ allows a detailed analysis of the structural reorganization involved in the conformational transition underlying transport. By aligning both structures on the scaffold domains, we observe the substrate binding site to elevate by 5.5 Å in the plane of the lipid bilayer during the transition from wide inward-open to the occluded conformation, in agreement with the elevator alternating access mechanism (Fig. 3B)[3]. During this movement, the scaffold domain only undergoes a minor rearrangement in TM6-TM7, which pivot ~7 Å laterally away from TM12 on the cytoplasmic side to accommodate for the substantial movement of TM1 towards the scaffold domain (Fig. 3A and Supplementary Fig. 9B).

Despite the significant translocation of the transport domain during the transition from $UraA_{WIO}$ to $UraA_{OCC}$, the relative positions of the interdomain-linkers with respect to the scaffold domain are

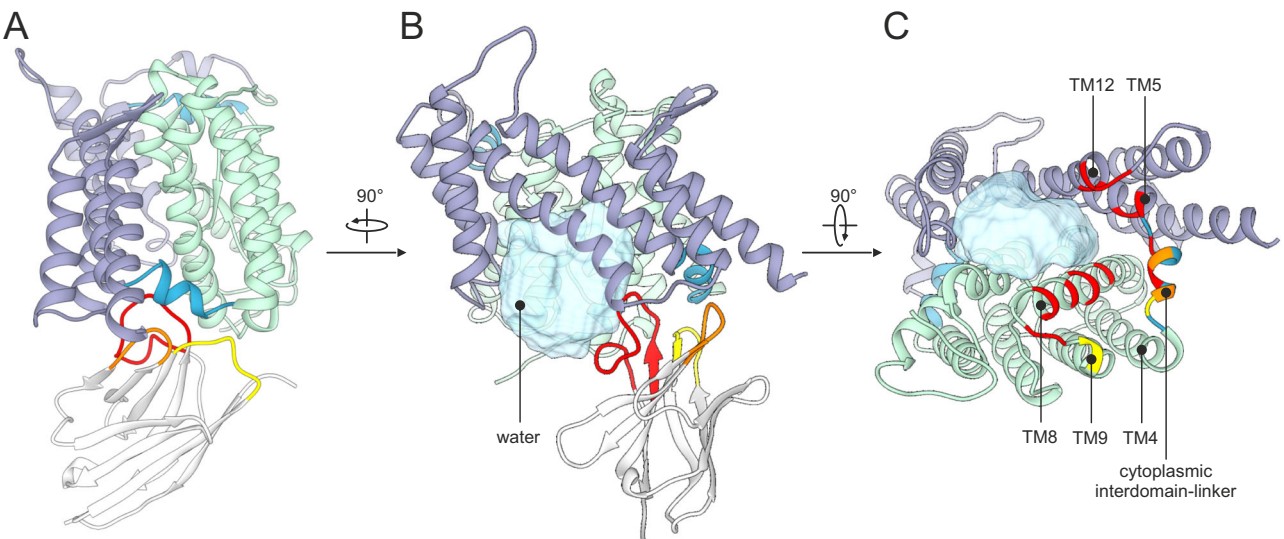

**Fig. 2 | Structural characterization of the Sy45 binding site. A** Structure of UraA(G320P)-Sy45 complex viewed from within the membrane. Transport and scaffold domain are purple and green, respectively. Interdomain-linkers are blue. Sy45 is colored gray and CDR1, CDR2, and CDR3 are yellow, orange, and red, respectively. **B** View of the UraA(G320P)-Sy45 complex in the plane of the membrane on the scaffold domain showing the access of cytoplasmic water to the substrate binding site. **C** View of UraA(G320P) from the cytoplasmic side. Sy45 has been left out for clarity. Regions of UraA within 4 Å of CDR1, CDR2, or CDR3 are colored yellow, orange, and red, respectively.

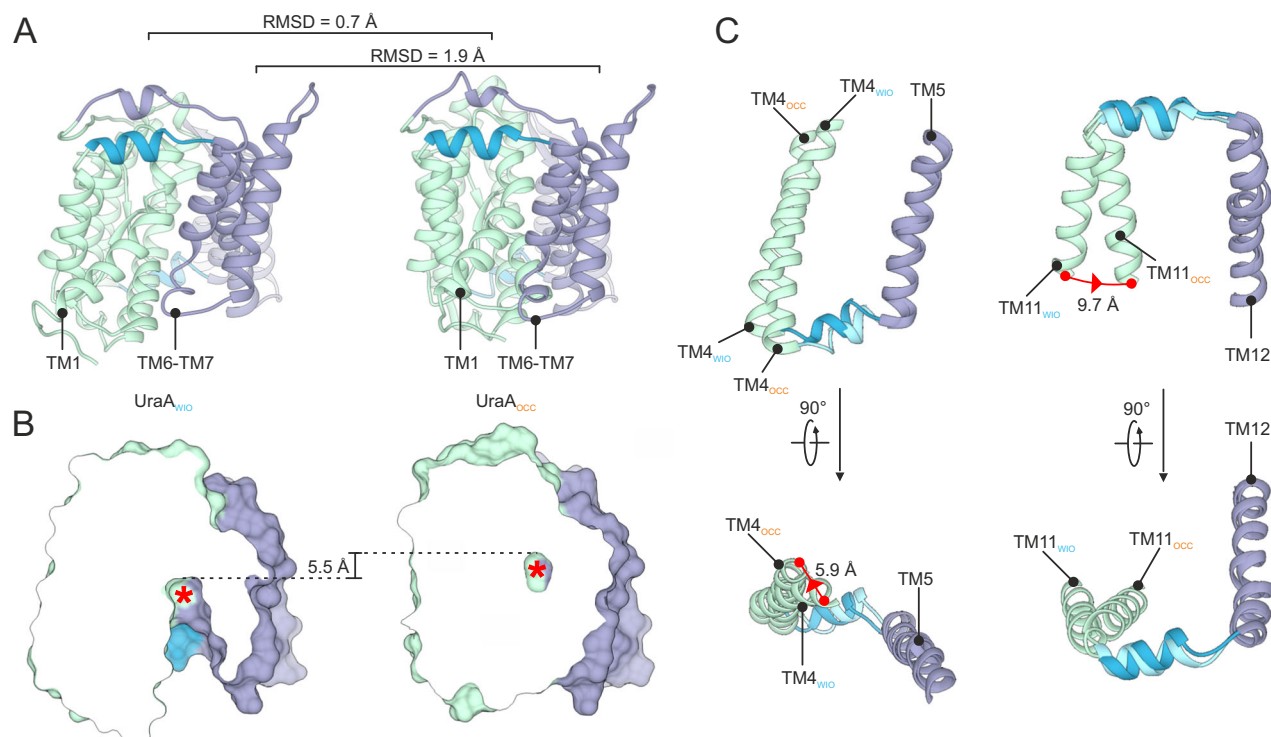

**Fig. 3 | Conformational transitions in UraA are mediated by interdomain linkers. A** Cartoon representation of single UraA$_{WIO}$ and UraA$_{OCC}$ protomers with scaffold and transport domain in green or purple, respectively, and the spacer helices in blue. **B** Surface representation of single UraA$_{WIO}$ and UraA$_{OCC}$ protomers clipped through the substrate binding sites. The location of the substrate is indicated by a red asterisk. **C** Conformational changes in TM4 to TM5 and the cytoplasmic interdomain-linker (left panel) and TM11 to TM12 and the external interdomain-linker (right panel). View is from within the plane of the membrane (top panels) or on the external side of the membrane (bottom panels). For superimposition of the structures TM5 and TM12 were used.

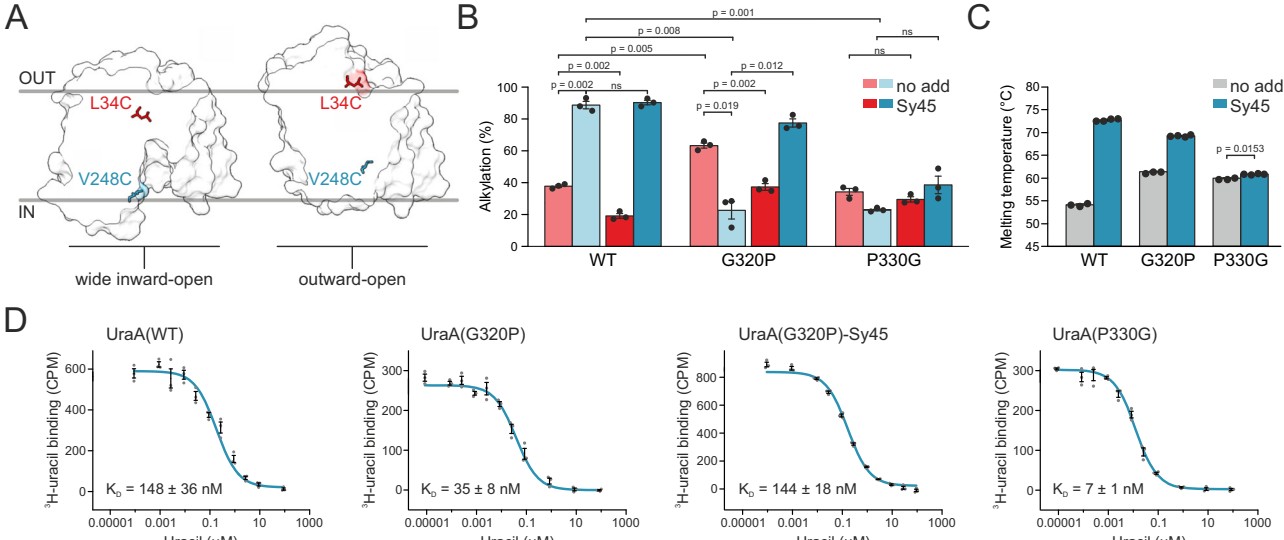

**Fig. 4 | Conformational equilibrium in UraA is affected by interdomain-linker mutants with an impact on substrate binding. A** The reporter cysteine mutants L34C and V248C are surface-exposed in an outward-open model or wide inward-open structure of UraA, respectively. **B** Degree of cysteine alkylation of L34C (red, outward-open reporter) and V248C (blue, wide inward-open reporter) in the presence and absence of Sy45. UraA variants were alkylated with mPEG5k for one hour and alkylation was quantified by mobility shift in SDS-PAGE and densitometry analysis. Data was normalized to the maximal alkylation obtained in the presence of 1% SDS. Shown are three technical replicates as scatter plot and derived mean values ± SEM as bars. A two-tailed, unpaired t-test was performed to test for statistical significance, The respective p-values are indicated. **C** Thermal stability of inter-domain linker mutants analyzed by differential scanning fluorimetry in absence (gray, $n = 3$) and presence (blue, $n = 4$) of Sy45 with technical replicates shown as scatter plot and derived mean values ± SEM as bars. A two-tailed, unpaired t-test was performed to test for statistical significance ($p = 0.0153$). **D** Scintillation proximity assay of UraA wild type, G320P and P330G in absence and presence of Sy45 as indicated with three technical replicates shown as gray scatter. Mean values and derived SEM shown as black error bars. Scintillation data was fitted in Origin with a binding curve for homologous competition to calculate the dissociation constants.

largely maintained, suggesting that functionally these regions are part of the latter (Fig. 3C). In the external linker the angle of the spacer helix with TM11 of the transport domain decreases by 17°. The hinge point for this rotation is located between Cys-318 and Val-319. In the cytoplasmic interdomain-linker, the angle between the spacer helix and TM4 of the transport domain slightly increases. These opposite changes in angles, combined with the small yaw movement of the cytoplasmic spacer helix that reduces the distance between TM4 and TM5, appear to underlie the full UraA$_{WIO}$ to UraA$_{OCC}$ transition.

**Interdomain-linker mutations affect conformational space**
To further elucidate the mechanistic relevance of the conserved glycine and proline residue in the external interdomain-linker, we analyzed the impact of the G320P and P330G substitutions on the conformational space of UraA. Initially, we determined the accessibility of substituted cysteines for a 5 kDa PEG-maleimide (mPEG5K)[38] in the presence and absence of our conformational probe Sy45. We engineered the single cysteine variants L34C and V248C to serve as reporters for the outward- and wide inward-open state, respectively (Fig. 4A). These positions were selected based on their accessibility either in our wide inward-open UraA structure or in an outward-open model[39] built using SLC4A1 as template (Supplementary Fig. 12)[10]. Leu-34 and Val-248 are not part of the Sy45 epitope nor surface-exposed in the occluded conformation. We chose positions in the transport domain rather than the scaffold domain to minimize any impact of changes in the oligomeric state on the conformational dynamics. The single cysteine variant L85C, which is well-exposed in every protein conformation, served as a positive control. All mutants were transport-competent, suggesting that all relevant conformations were sampled (Supplementary Fig. 13).

Wild type UraA showed a two-fold higher degree of labeling with mPEG5K for the wide inward-open reporter compared to the outward-open reporter (Fig. 4B and Supplementary Fig. 14). Addition of the

conformational probe Sy45 reduced labeling of the outward-open reporter due to the shifted conformational equilibrium towards the wide inward-open state. However, a concomitant increase in the degree of alkylation of the wide inward-open sensor was not observed. Similar observations were made for UraA(G112P) (Supplementary Fig. 14), indicating that the conformational equilibrium of this mutant is not substantially altered.

For UraA(G320P), we observed more alkylation of the outward-open sensor and significantly reduced accessibility of the wide inward-open sensor compared to wild-type UraA (Fig. 4B), suggesting an altered conformational equilibrium. Consistently, in the presence of Sy45 the population of UraA(G320P) occupying the wide inward-open state increased, indicating that the mutation did not arrest the protein in a specific conformation. Notably, the apparent change in conformational space occupancy was not caused by increased self-association of UraA(G320P) (Fig. 1E, Supplementary Fig. 4B, C, and Supplementary Table 1), as separately isolated UraA(G320P) monomer and dimer fractions showed very similar cysteine accessibility profiles (Supplementary Fig. 14). The notion that UraA(G320P) populates a different section of the transporter's conformational space is further supported by its four-fold higher uracil binding affinity (Fig. 4D). Again, addition of Sy45 reverted the variant back to a state with an affinity very similar to wild type UraA[13].

In contrast to wild-type UraA and UraA(G320P), we observed only little mPEG5K labeling of either the outward- or inward-facing sensor for UraA(P330G) (Fig. 4B). We interpret these results as a low occupancy of the corresponding conformations in UraA(P330G), suggesting a high population of the occluded state. The addition of Sy45 did not significantly affect the conformational equilibrium (Fig. 4B), nor increased the melting temperature of the mutant substantially (Fig. 4C, Supplementary Fig. 6), suggesting a strongly reduced binding efficiency of Sy45 to UraA(P330G). This underlines that the P330G mutation constrains the transporter to a conformational space no longer sampling the wide inward-open state. However, UraA(P330G) is still able to

expose its binding site to the bulk solvent as we do observe substrate binding (Fig. 4D). The twenty-fold increase in binding affinity suggests that following substrate binding, the transporter spends a comparably long time in an occluded state thereby preventing uracil dissociation.

We further analyzed the impact of the G320P and P330G mutations on the protein conformational dynamics more extensively and at a higher spatial resolution using hydrogen-deuterium exchange mass spectrometry (HDX-MS). For all UraA variants we reached a sequence coverage between 81.5% (wild type UraA) and 90.2% (UraA(G320P)) with an average redundancy of 4.91 peptides per amino acid (Supplementary Fig. 15). The overall rate of deuterium incorporation showed a high correlation with the degree of solvent exposure expected based on the protein structure (Supplementary Fig. 16 and 18), indicating that the structural integrity of UraA was maintained throughout the experiments.

For wild-type UraA, the addition of $100\,\mu M$ uracil induced a decrease in deuterium uptake in the substrate binding site (TM3 and TM10) and adjacent regions (TM1 and TM8) (Fig. 5A and Supplementary Fig. 17). These findings suggest an increased population of the

occluded state upon uracil binding. An overall similar reduction in HDX in the presence of substrate is observed for UraA(G320P) (Fig. 5B), apart from the extracellular side of TM1 that showed a more reduced deuterium uptake. The deuterium uptake of UraA(P330G) was barely affected upon the addition of substrate (Fig. 5B), despite its very high affinity for uracil (Fig. 4D). This result further strengthens our hypothesis that P330G constrains the transporter by severely constricting its conformational space.

Changes in conformational dynamics are best visualized by directly comparing the relative deuterium uptake between the wild-type protein and the mutants in their apo states (Fig. 5C). UraA(G320P) showed a decrease in HDX on the cytoplasmic side of TM8 in agreement with our alkylation analysis (Fig. 4B). While the differential deuterium uptake observed for UraA(P330G) shows similar agreement with our cysteine accessibility studies, the increased spatial resolution of HDX-MS highlights that deuterium uptake is significantly reduced in almost the entire side of the transport domain facing the scaffold domain (Fig. 5C). A decrease in uptake was observed on the extracellular side of TM1, consistent with the assumption that this variant

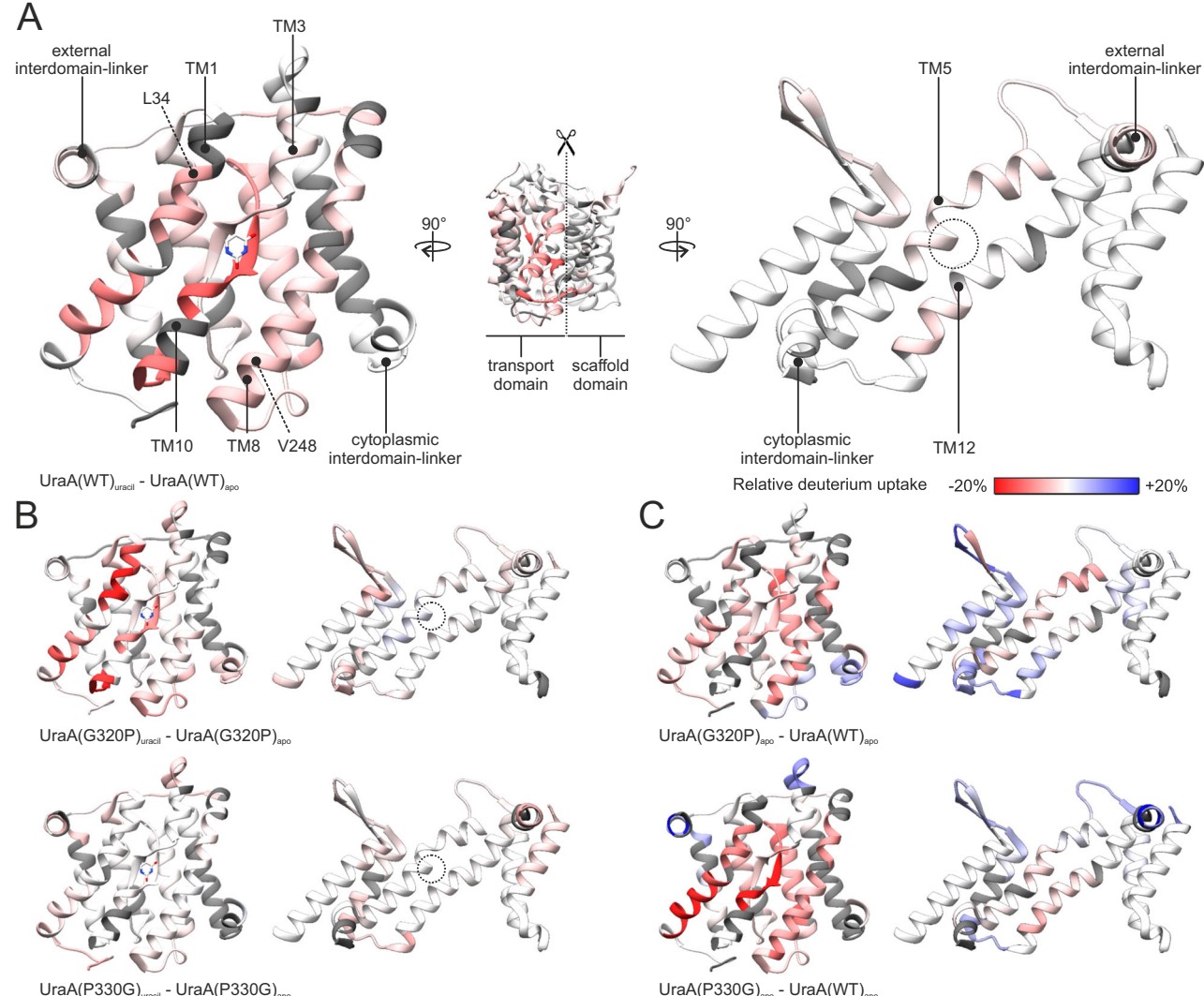

**Fig. 5 | Hydrogen-deuterium exchange mass spectrometry analysis reveals changes in the conformational dynamics of UraA. A** HDX of wild type UraA in the presence and absence of $100\,\mu M$ uracil. Transport and scaffold domain are oriented as if the protein was folded open as indicated in the central panel. The interdomain-linkers are shown in both domains as a reference point. The circle on the scaffold domain represents the position of the substrate in the opposing transport domain assuming the $UraA_{OCC}$ conformation. **B** Substrate-dependent HDX for UraA(G320P) and UraA(P330G). **C** Differential HDX resulting from the G320P (top panel) and P330G (lower panel) mutations in the absence of ligand compared to wild type UraA. The color range representing the relative deuterium uptake, ranging from red (decreased uptake) over white (no change) to blue (increased deuterium uptake), applies to all panels. Gray stretches indicate regions of not identified amino acids.

adopts an occluded conformation. Although changes in deuterium uptake in the scaffold domain helices TM5 and TM12 appear to be consistent with altered conformational space occupancy in UraA(G320P) and UraA(P330G), we refrain from a detailed analysis of these regions due to the unclear consequences on the solvent accessibility of the different monomer-dimer equilibrium observed for these mutants (Fig. 1E).

## Discussion

Thus far the mechanistic relevance of interdomain-linkers in elevator transport proteins is only implicitly acknowledged by the many structures highlighting rigid body movements as the essence of this transport mode. Here, by manipulating the flexibility of interdomain-linkers in the SLC23 transporter UraA, we directly demonstrate functionality beyond that expected for mere tethers. We specifically focused on reverting the flexibility or rigidity of the conserved amino acids in the loops flanking the spacer helix in the external interdomain-linker. The respective mutants UraA(G320P) and UraA(P330G) displayed excellent biochemical behavior, formed dimers, and bound substrate, yet did not demonstrate significant uracil transport rates under the tested conditions (Fig. 1). Our data allows us to propose that changes in the conformational space of these mutant transporters underlie this reduction in transport activity.

Both cysteine-alkylation and HDX-MS, which provides a higher spatial resolution, suggest that the conformational equilibrium of unliganded UraA(G320P) has shifted away from the mostly inward-open conformation observed for the wild type. The decreased surface exposure of the cytoplasmic cavity and the reciprocal increased accessibility of the external cavity suggest a higher occupation of an outward-open conformation (Figs. 4B and 5C). Using our conformational probe Sy45, we further demonstrated that UraA(G320P) is not arrested in this conformation and that its conformational equilibrium can be shifted towards UraA$_{WIO}$ (Fig. 4B), in agreement with the fact that other SLC23 subfamilies holding a native proline at this approximate position are capable of transport[40] (Supplementary Fig. 2). Within the limits of our experimental resolution, all transport-relevant conformations thus appear to be accessible. Assuming that our data on detergent-solubilized protein are representative of the behavior of the protein within the confines of the lipid membrane, it seems reasonable to propose that not the conformational space itself, but its occupancy is altered for UraA(G320P) (Fig. 6A). This may lead to a reduced frequency of reaching one of the states required to complete the transport cycle, thereby preventing reliable detection of transport.

Mutation of Pro-330, a residue strongly conserved throughout the 7TMIR architecture (Supplementary Fig. 3), appears to affect the conformational space of the transporter differently. Exposure of both

the internal and external cavities in UraA(P330G) were strongly reduced (Figs. 4B and 5C), suggesting that this mutant preferentially assumes an occluded conformation. Attempts to stabilize the mutant in different conformations by the conformational probe Sy45 or substrate were unsuccessful. Nevertheless, given that substrate can still bind (Figs. 1F and 4D), UraA(P330G) is not arrested in an occluded conformation but still occasionally samples at least one open conformation. Together, the strongly reduced exposure of internal surfaces and the lack of deformability suggest that UraA(P330G) almost exclusively occupies the section of conformational space assigned to the occluded state. Other conformers are sampled with very low frequencies (Fig. 6A). The strong impact of the P330G mutation on the conformational space of UraA is in agreement with a recent molecular dynamics simulation of the cyanobacterial bicarbonate transporter BicA[41], whose membrane domain has the same 7TMIR-fold as UraA. Glycine substitution of Pro-122 and Pro-341 in BicA, equivalent to Pro-121 and Pro-330 in UraA, altered the conformation free-energy landscape of the transporter and generated an additional free-energy barrier reducing the transition from the occluded state.

Both UraA(G320P) and UraA(P330G) showed a significant increased affinity for uracil, amounting to a twenty-fold increase for the latter, despite the remote location of these mutations with respect to the substrate binding site. Similar observations were previously made for MBP, the soluble maltose-binding protein from *E. coli*[42]. While mutations in the hinge region connecting both MBP domains maintained the ligand-binding interface, the maltose binding affinity was increased up to fifty-fold because of reduced intrinsic opening rates. In essence, the closed form of MBP enclosed the ligand thereby preventing its dissociation. We suspect that the moderate to strong increase in uracil binding affinity observed for our mutants relies on a similar principle and reflects an increased occupancy of the occluded state.

UraA(G320P) in complex with Sy45 crystallized in a wide inward-open structure. The high similarity of the scaffold domain in UraA$_{WIO}$ and UraA$_{OCC}$ suggests that the structural differences with UraA$_{IO}$, in particular the large apparent rearrangement of TM6 and TM7, are not part of the linear transport mechanism as suspected previously[13]. However, the scaffold domain does not appear to be as rigid as the transport domain. We observe a small shift of the cytoplasmic side of TM6-TM7 to the periphery of the dimer that provides space for TM1 in the transport domain to close the inside cavity during the UraA$_{WIO}$-UraA$_{OCC}$ transition. Similar subtle adaptations of the scaffold domain in response to mutation-induced changes in the conformational landscape of the transport domain may contribute to the enhanced stability of the UraA(G320P) and UraA(P330G) dimers. Furthermore, the increased dimer stability among our mutants with decreased

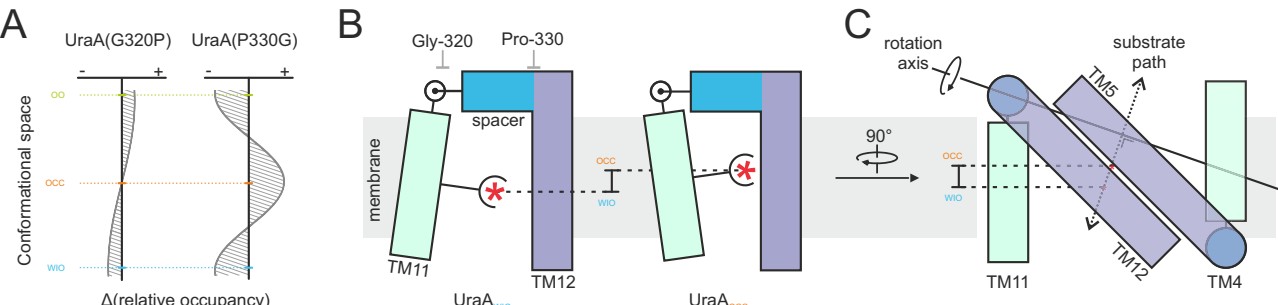

**Fig. 6 | Role of the interdomain-linker in the transport mechanics. A** Theoretical profile of the mutation-induced changes in the occupancy of the conformational space relative to wild type UraA. WIO, OCC, and OO refer to the wide inward-open, occluded, and outward-open conformations, respectively. Positive or negative deviations indicate increased or reduced occupancy of the respective conformation. **B** Toy model of the external interdomain-linker demonstrating how the

conversion from UraA$_{WIO}$ to UraA$_{OCC}$ results in a change in the vertical position of the substrate. View is in the plane of the membrane. Substrate is represented by a red asterisk. **C** Model of TM4-5 and TM11-12 highlighting the approximate location of the rotation axis[66] for the transport domain and its consequences on the trajectory of the substrate with respect to the barrier helices TM 5 and 12. View is in the membrane on the scaffold domain.

conformational flexibility suggests some degree of conformational 'breathing' in the scaffold domain during transport, in line with the reduced transport rates observed upon cross-linking the scaffold domains of the SLC13 transporter VcINDY[43].

The fixed positions of the spacer helices relative to the scaffold domain serve to place the position of the hinge point, around which the transport domain pivots, behind the plane of the substrate binding site (Fig. 6B). Thereby, comparably small, local conformational changes in the hinge point result into elevation of the transport domain and a vertical displacement of the substrate. The position of the rotation axis that describes the movement of the transport domain during the UraA$_{WIO}$-UraA$_{OCC}$ transition is approximately parallel to TM5 and TM12 in the scaffold domain. These helices constitute the fixed-barrier against which the substrate binding site in the transport domain moves to translocate the substrate to the other side[3]. This specific orientation of the rotation axis results in a displacement vector of the substrate binding site that is nearly orthogonal to the barrier helices TM5 and TM12 (Fig. 6C). As a result, substrate transport follows the shortest translocation path possible. In combination with the need for only small local changes in the protein backbone to enable transport, this may contribute to the exceptionally high channel-like transport rates observed for other transporters with this fold, e.g. human SLC4A1[44,45], SLC26A5[46], and SLC26A9[20].

Short interdomain-linkers containing a spacer helix are commonly observed in elevator proteins[7] (Supplementary Fig. 1). As observed here and previously for the SLC13 and 2HCT families[31,47], they maintain a defined orientation to the scaffold domain during conformational transitions but show flexibility on the side of the transport domain. In the SLC23, SLC4, and SLC26 families this rigidity with respect to the scaffold domain results from a proline, particularly conserved in the external interdomain-linker (Supplementary Fig. 3). Alternatively, rigidity in the SLC13 family is achieved by conserved stretches of bulky residues (external interdomain-linker) or a conserved salt bridge on the cytoplasmic side[47]. The fixed orientation of the interdomain-linker with respect to the scaffold domain appears of general relevance. Neutralization of one of the residues constituting this salt bridge in the scaffold domain of NaDC-1 abolishes transport[48], similar to our observations for UraA(P330G). Furthermore, disruption of this salt bridge by an alternative interaction of the respective arginine with a glutamate in the STAS domain of SLC26a6 was proposed to underlie the inhibitory effect of the latter. Together, this illustrates the wide-spread mechanistic relevance of interdomain-linkers in elevator-type transport proteins.

Solute carriers have recently gained recognition as a promising class of drug targets[1,49,50]. While various drugs were designed to inhibit SLCs, it is primarily loss-of-function mutations in human SLCs that are associated to Mendelian diseases, underscoring the need for positive allosteric modulators[51]. Based on the current work we conclude that enhancing the flexibility and rigidity of the respective regions in the interdomain-linkers may contribute to this type of direct pharmacological potentiation of elevator-type transporters.

## Methods

### Molecular cloning
The open reading frame of UraA was amplified from genomic DNA of *E. coli* MC1061[52]. Primers were automatically designed online (https://www.fxcloning.org). Primer sequences are provided as Supplementary Data 1. A gel-purified PCR product was cloned into pINIT_cat using FX cloning[53]. Mutants of *uraA* were generated by megaprimer PCR. Inserts were sequence-verified, subcloned into the pBXC3GH or pBXCA3GH vector, and transformed into *E. coli* MC1061.

### Protein expression and purification
*E. coli* MC1061 containing pBXC3GH-UraA or variants thereof was cultivated at 37 °C in a fermenter (Bioengineering) containing TB medium supplemented with 100 µg/mL ampicillin and 1 mM MgSO$_4$. At an OD$_{600}$ of 1.5 the temperature was gradually decreased to 25 °C over the course of one hour. Expression was induced with $1 \times 10^{-2}$% (w/v) L-arabinose[54] and continued for 16 h. Cells were resuspended in 20 mM Hepes pH 7.5, 150 mM NaCl, 1 mM MgCl$_2$, 1 mg/ml lysozyme, 20 µg/ml DNAseI and stirred for 1 h at 4 °C before disruption using a Menton-Gaulin homogenizer (APV) operated at 400 bar in presence of 1 mM PMSF. Cell debris was removed by centrifugation at 15,000×g and 4 °C for 30 min. Membrane vesicles were collected by centrifugation at 140,000 × g for 1 h at 4 °C and the pellet was resuspended in purification-buffer (20 mM Hepes pH 7.5, 150 mM NaCl, 10% glycerol (v/v)) using a Potter-Elvehjem homogenizer. Resuspended vesicles were snap-frozen and stored at −80 °C until use. Protein purification was performed at 4 °C. 10 g membrane vesicles were thawed and solubilized for 1 h in 100 ml purification-buffer supplemented with 1.5 g n-decyl-β-D-maltopyranoside (Glycon Biochemicals GmbH) and 15 mM imidazole. After 30 min centrifugation at 140,000 × g the supernatant was submitted to batch binding with washed and pre-equilibrated Ni-NTA resin for 1 h. The sample was loaded on a gravity flow column. Following draining of the column, the resin was washed with 20 column volumes purification buffer supplemented with 0.2% (w/v) DM and 50 mM imidazole. UraA was eluted from the column by a 30 min incubation under mild shaking with 1 CV purification-buffer supplemented with 1.2 mg HRV-3C protease[55] and 0.2% (w/v) DM. The column was drained and washed with 3 CV purification-buffer supplemented with 0.2% (w/v) DM. All fractions were pooled and concentrated using an Amicon Ultra 50 kDa MWCO concentrator. The concentrated sample was centrifuged for 10 min at 13,000×g to remove large aggregates and subsequently loaded on a Superdex 200 increase 10/300 GL size exclusion column equilibrated with purification-buffer supplemented with 0.2% (w/v) DM or 10 mM Hepes pH 7.5, 150 mM NaCl, 0.2% DM (w/v) in case the sample was used for crystallization. Peak fractions (UraA(WT) and UraA(G112P): monomeric fraction; UraA(P330G): dimeric fraction; UraA(G320P): monomeric and dimeric fractions) were pooled and concentrated. Protein for the selection of synthetic nanobodies and SPA-based binding assays was expressed with a C-terminal AVI-tag and enzymatically biotinylated[56]. Biotinylation efficiency was quantified using the mobility shift of biotinylated protein in SDS-PAGE upon the addition of Streptavidin and exceeded 90% for all samples. Synthetic nanobodies were produced in *E. coli* MC1061 from a pSB_init-derivative containing the open reading frame coding for the respective sybody[57]. Cells were cultivated at 37 °C in TB medium supplemented with 34 µg/mL chloramphenicol and 1 mM MgSO$_4$. At an OD$_{600}$ of 1.0 the temperature was gradually decreased to 25 °C over the course of one hour. Expression was induced with $1 \times 10^{-2}$% (w/v) L-arabinose and continued for 16 h. Cells were harvested by centrifugation for 15 min at 15,000 × g and 4 °C. Cell pellets were resuspended in 50 mM potassium phosphate, pH 7.5, 300 mM NaCl, 20 µg/ml DNAseI, 1 mg/ml lysozyme, 15 mM imidazole, and 1 mM PMSF. Following a 2 h incubation at 4 °C, the sample was disrupted with a Stansted Homogenizer EP (Stansted Fluid Power LTD) and subsequently centrifuged for 15 min at 15,000 × g and 4 °C. The sample was loaded on a gravity flow column and following draining, the column was washed with 20 column volumes 50 mM potassium phosphate, pH 7.5, 300 mM NaCl, 50 mM imidazole, and subsequently eluted with 50 mM potassium phosphate, pH 7.5, 300 mM NaCl, and 300 mM imidazole. for 30 min at 15,000×g and 4 °C. The supernatant was supplemented with Ni-NTA resin and incubated for 1 h. Concentrated peak fractions were loaded on a Sepax SRT-10C-SEC-300 size exclusion column equilibrated with 10 mM Hepes, pH 7.5, and 150 mM NaCl, in case the sample was used for crystallization. Peak fractions were pooled, concentrated, flash-frozen in liquid nitrogen and stored at −80 °C until use.

### Sy45 selection
Sy45, which displays a convex paratope, was selected against wild type UraA as described[34,57]. In short, one round of ribosome display

followed by two rounds of phage display were performed for each of the three sybody libraries. Binders were selected based on their ELISA-signal intensity and sequence analyzed.

## Protein crystallization

UraA-Sy45 protein complexes were prepared by mixing SEC-purified UraA and Sy45, supplemented with 0.2% (w/v) DM, at a molar ratio of 1:1.2 and a final protein concentration of 10 mg/ml. The sample was supplemented with 1% NG (w/v) and incubated for at least 30 min. Crystals of UraA(G320P)-Sy45 were grown in sitting drops at 18 °C by vapor diffusion after mixture of protein and reservoir solutions in a 1:1 ratio. Best crystals were obtained using a reservoir solution containing 50 mM Tris.HCl, pH 8.4, 50 mM magnesium acetate, 35% (v/v) PEG400, 0.1% (w/v) benzamidine hydrochloride. Crystals of UraA(G320P)-Sy45 containing uracil were generated using a reservoir solution containing 50 mM Tris.HCl, pH 8.4, 50 mM magnesium acetate, 40% (v/v) PEG400, 0.1% (w/v) benzamidine hydrochloride and 1 mM uracil. Apo crystals were used as crystal seeds. Crystals were fished with nylon loops and flash-frozen in liquid nitrogen.

## Structure determination

The diffraction data yielding the UraA(G320P)-Sy45 co-crystal structure was collected at the P13 beamline of PETRA III[58] at a wavelength of 0.98 Å and processed using XDS and scaled using XSCALE from the XDS package[59]. The data was corrected for anisotropy with the UCLA diffraction anisotropy server[60]. The structure was solved by molecular replacement using Phaser-MR from the Phenix suite[61] with the occluded UraA structure[13] and a SWISS-MODEL[39] of the synthetic nanobody Sy45 as model. The structure was refined with Phenix.refine at 3.5 Å resolution[61]. The dataset for the uracil-liganded UraA(G320P)-Sy45 structure was collected at the X06DA beamline of the Swiss Light Source (SLS) at a wavelength of 1.00 Å. The dataset was processed and corrected for anisotropy as described for the apo-UraA(G320P)-Sy45 structure. The structure was solved by molecular replacement using the apo-UraA(G320P)-Sy45 structure and refined at 3.7 Å resolution.

## Radioisotope transport assays

Transport studies were performed in *E. coli* BW25113 *ΔuraA*[62]. Cells were transformed with the pBXC3GH plasmid carrying the sequence of the respective UraA variant. Cells were grown in TB medium containing 100 µg/ml ampicillin, 50 µg/ml kanamycin, and 1 mM MgCl$_2$ at 37 °C to an OD$_{600}$ of 1.0 to 1.5. Gene expression was induced with $1 \times 10^{-4}$% (w/v) L-arabinose (w/v) and cultivation continued for 1 h. Cells were washed with ice-cold buffer containing 50 mM potassium phosphate, pH 7.2, 1 mM MgCl$_2$ and resuspended to an OD$_{600}$ of 24 in uptake buffer containing 50 mM potassium phosphate, pH 6.5, 1 mM MgCl$_2$, 0.2% D-glucose (w/v), and stored on ice. Prior to the transport assay, the cell suspension was diluted 20-fold in uptake buffer, and equilibrated for 2 min at 25 °C while stirring. Uracil uptake was initiated by the addition of 250 nM [5,6-$^3$H]-uracil (ARC-USA) and stopped at regular intervals by 20-fold dilution of a 100 µl aliquot in ice-cold uptake buffer followed by rapid filtration on 0.45 µm nitrocellulose filters (Sartorius AG). After washing the filters with another 2 ml buffer, the radioactivity associated with the filter was determined by scintillation counting using a Hidex 300SL (Hidex). For determining the protein expression levels, the cell pellet from 400 µl cell suspension was resuspended in 200 µL 20 mM Hepes, pH 7.5, 150 mM NaCl, 1 mM MgCl$_2$, 20 µg/mL DNAseI, 1 mM PMSF and supplemented with 300 mg glass beads with a diameter of 100 nm. Cells were disrupted by two runs in a bead-beater for 20 sec at 4.0 m/s and incubation for 5 min on ice in between. Following sedimentation of the beads, the cell lysate was supplemented with 5x SDS-PAGE sample buffer and additionally supplemented with 2% (w/v) SDS. The sample was analyzed by SDS-PAGE and UraA-GFP constructs were detected by in-gel fluorescence

detection with an Image Quant LAS 4000 with excitation wavelength of 460 nm and Y515 filter.

## SPA-based binding assay

Streptavidin PVT scintillation beads (Perkin Elmer) were diluted to 10 mg/ml in 20 mM Hepes, pH 7.5, 150 mM NaCl, 0.2% DM (w/v). Bio-tinylated UraA-variants were incubated with $^3$H-uracil (150 nM wild type UraA with 34 nM $^3$H-uracil; 50 nM UraA(G320P) with 7 nM $^3$H-uracil; 7 nM UraA(P330G) with 7 nM $^3$H-uracil) and increasing amounts of unlabeled uracil. Samples containing 5 µM Sy45 were prepared with 150 nM wild-type UraA and 34 nM $^3$H-uracil, 150 nM UraA(G320P) and 34 nM $^3$H-uracil, or 50 nM UraA(P330G) and 7 nM $^3$H-uracil. Samples were equilibrated for 2 h at room temperature to reach equilibrium. Each sample was mixed with 170 µg Streptavidin PVT scintillation beads in one well of a 96-well plate and incubated for 1 h at room temperature in the dark. Scintillation was recorded using a plate-reader (Perkin Elmer MicroBeta Trilux 1450 LSC). Background signal was recorded after 1 h incubation in the presence of 1% (w/v) SDS and subtracted from each data point. Mean and standard error were calculated from triplicates, plotted against the concentration of unlabeled uracil and fitted in Origin using Eq. (1) to calculate the binding affinity.

$$\text{binding (CPM)} = \frac{\text{max. binding (CPM)} \times [^{14}C - \text{uracil}]}{[^{14}C - \text{uracil}] + [^{12}C - \text{uracil}] + K_D} \\ + \text{background (CPM)} \quad (1)$$

## Cysteine alkylation assay

Single cysteine variants of UraA were purified in the presence of 5 mM beta-mercapto-ethanol (bME) throughout the whole purification protocol starting from cell disruption. Prior to alkylation, bME was removed from the sample using a Biospin-6 column equilibrated with labeling buffer containing 20 mM Hepes, pH 7.5, 150 mM NaCl, 10% glycerol (v/v), 0.2% DM (w/v) and the eluted protein was diluted to 12 µM by addition of labeling buffer. Samples containing Sy45 were supplemented to a final concentration of 50 µM Sy45. Samples were equilibrated for 10 min, and alkylation was started by the addition of 1 mM 5 kDa PEG-maleimide (mPEG5K). Alkylation was quenched at different time points by addition of 5x SDS-PAGE sample buffer containing 100 mM DTT. The zero time point sample was prepared without the addition of mPEG5k and the control sample was alkylated for 1 h in presence of 1% SDS (w/v). All samples were loaded on 12% SDS-PAGE and subsequently stained with Coomassie brilliant blue R-250. Alkylation was quantified by densitometry analysis using ImageJ and the labeling efficiency calculated for each time point using Eq. 2 with St being the band intensity of the actual sample at a given time point and C being the control

$$\text{labeling efficiency (\%)}$$
$$= \frac{\text{labeled band St}/(\text{labeled band St} + \text{unlabeled band St})}{\text{labeled band C}/(\text{labeled band C} + \text{unlabeled band C})} \times 100 \quad (2)$$

## Differential scanning fluorometry

UraA variants were diluted to 0.1 mg/mL in 20 mM Hepes pH 7.5, 150 mM NaCl, 0.2% (w/v) DM. Measurements in presence of uracil were performed by addition of 1 mM uracil. Samples were equilibrated for 30 min at 4 °C. The thiol-reactive coumarin 7-diethylamino-3-(4′-maleimidylphenyl)-4-methylcoumarin (CPM) was added to a final concentration of 40 µM and samples were equilibrated for 5 min at room temperature. After centrifuged at 13,000 × g and 4 °C for 10 min, the remaining supernatant was subjected to thermal melting from 25 to 90 °C with a ramp of 2 °C/min using the Rotor-Gene Q (Qiagen). CPM was excited at 365 ± 20 nm and fluorescence was detected at

460 ± 20 nm with a detector gain setting of −2. The first derivative of the melting curve showing the melting temperature as the local maximum was generated using the Rotor-Gene Q software.

## HDX sample preparation

UraA variants were expressed and purified as described above. Size exclusion chromatography elution fractions corresponding to monomeric and dimeric forms of the respective UraA variant were pooled. Protein samples were stored at −80 °C and freshly thawed for each experiment. Before each HDX-MS experiment, labeling (L)-, equilibration (E)- and quench (Q)- buffers were freshly prepared with $D_2O$ or $H_2O$, respectively (L/E-buffer: 20 mM Hepes, 150 mM NaCl, 0.2% DM at pH/D 7.5 containing 100 µM of the respective ligand (uracil, binding) or without any ligand (control); Q-buffer: 150 mM potassium phosphate buffer, 0.2% (w/v) DM at pH 2.20). The final sample concentration of 0.675 mg/mL, which equates to 25 pmol on pepsin column, was generated by diluting the protein stock solution with its corresponding E-buffer. Next, buffers and samples were equilibrated and stored at 20 °C, for E- and L-buffer, and at 0 °C for Q-buffer and protein sample.

## HDX mass spectrometry

For HDX-MS experiment a fully automated HDX-2 system (Waters, Milford USA) was used. For each condition, data were acquired for four technical replicates. After equilibration of the sample and all the buffers, the experiment was started by diluting 4 µL of sample in 56 µL (resulting in a protein concentration of 0.045 mg/ml) of either E-buffer, for non-deuterated reference measurements, or in L-buffer for five time points (0, 30, 360, 900 and 2700 sec of labeling). Next, the 1 to 15 diluted sample was quenched by diluting 50 µL sample in 50 µL ice-cold Q-buffer. After 0.5 min of quenching, 95 µL sample was injected into the temperature-controlled chromatography system (HDX nanoAqcuity UPLC, Waters) equipped with a 50 µL sample loop, for online digestion and reverse-phased chromatographic separation of the generated peptides. Online digestion was carried out on a pepsin column (Enzymate BEH pepsin column; 2.1 × 30 mm; Waters) for 3 min at 100 µL/min with 100% phase A ($H_2O$ with 0.23% FA). Next, the generated and eluting peptides where trapped and washed on a C18 pre-column (C18 1.7 µM VanGuard 2.1 × 5 mm pre-column; Waters). After digestion and trapping the peptides were separated on a reverse phased column with a linear gradient ranging from 5% phase B (ACN with 0.23% FA) to 40% ACN with 0.23% FA for 7 min and 40 µL/min. Next, a rapid rise to 95% B for 2 min followed by a washing step at 95% B for 2 min and an equilibration step at 5% B for 2 min. The chromatographic separation was kept at 0 °C to reduce back exchange. Eluting peptides were directly infused into the Synapt G2-Si mass spectrometer operated in HDMS$^E$ mode (50 - 2000 $m/z$). This mode utilizes ion mobility separation (IMS) to add another dimension to the peptide separation (RT, IMS, $m/z$). The mass spectrometer is equipped with a Z-spray ESI source with an additional independent lockmass sprayer for lockmass infusion (GluFib, 785.8426 $m/z$).

## HDX data evaluation and statistics

For peptide identification, non-deuterated peptides for each condition (control and binding) were subjected to Protein Lynx Global Server 3.0.3. (PLGS, Waters) software. Peptides with a high confidence score (over 6) and identified in three out of four replicates were retained for further evaluation. For peak picking of all corresponding peaks through calculating weighted average $m/z$ (centroid) for all isotopic peaks in each peptide DynamX 3.0. (Waters) was used (Supplementary Table 3). For relative deuterium uptake calculation in each peptide the centroid mass of deuterated isotope peaks were compared to their respective undeuterated reference peaks. This relative uptake was plotted against their respective labeling times and all assigned spectra were manually inspected and revised as necessary.

After manual inspection, all peptides (precursor) were subjected to a two-stage t-test previously described[63] to test for statistically significant uptake differences for all peptides. First, the average standard error of the mean (SEM) was calculated for each precursor by dividing the SEM across the quadruplicates in every timepoint and condition through the total number of time points. Next, the average SEM of each precursor was multiplied by the t-distribution value for a confidence interval of 95% (p ≤ 0.05%, two-sided, unpaired) with the number of replicates minus one as degrees of freedom, resulting in a test value for each precursor. If the absolute value for the deuterium uptake difference in each time point is equal or greater as the test value, then the time point is considered to show statistical significance. The precursor is submitted to the next stage if a majority (3 out of 4) label timepoints is accounted as significant. In the second stage, we calculated the sum of all uptake differences for each precursor and estimate the summed SEM of this summed uptake differences by multiplying the SEM of a precursor with the number of time points. As t-distribution value, we use 99% as confidence interval (p ≤ 0.01%, two-sided, unpaired) with the number of time points minus 1 as degrees of freedom. The t-distribution value is multiplied with the summed SEM of the precursor to form a second precursor specific test value. If the absolute value of the summed uptake differences of a precursor is equal or greater as the second test value, this precursor is accounted to show statistically significance. For further evaluation, only peptides were used which passed both stages of the t-test.

For direct comparison of apo-UraA(WT) and apo-UraA(mutant), we used the control condition from DynamX (without ligand) in each experiment and compared apo-UraA(mutant), labeled as 'binding', minus apo-UraA(WT), labeled as 'control'. Subsequently, these data were analysed using our established statistics pipeline. For the visualization of the missing amino acids, we used the missing amino acids from both experiments and combined them in the final figure using UCSF Chimera 1.16[64].

Supplementary Tables 4–6 provide HDX summary tables and Supplementary Data 2-6 provide HDX data tables formatted according to the recommendations of the HDX-MS community[65].

## Reporting summary

Further information on research design is available in the Nature Portfolio Reporting Summary linked to this article.

## Data availability

The source data underlying Figs. 1D, F and 4B–D, and Supplementary Figs. 13A and 14 are provided as Source Data File. The atomic coordinates have been deposited in the Protein Data Bank under accession code 8OMZ (unliganded UraA(G320P)-Sy45) and 8OO1 (liganded UraA(G320P)-Sy45). The previously published UraA structure in an occluded conformation is accessible under the accession code 5XLS. HDX-MS data tables are provided as Supplementary Data 2–6. The complete HDX-MS dataset including raw spectra has been deposited to the ProteomeXchange Consortium via the PRIDE partner repository with the dataset identifier PXD054394. Source data are provided with this paper.

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

## Acknowledgements

We thank the staff of the P13 and X06DA beamlines for support during data collection; Y. Thielmann and B. Rathmann for their support with crystallization screening; and K. M. Pos, H. R. Sikkema, I. Tascón, H. Michel, and N. Schenck for discussions. We acknowledge A. R. Mehdipour, L. Tesmer, G. Hummer, and M. Marass for stimulating discussions and corrections to the manuscript. This research was supported by the German Research Foundation via the Cluster of Excellence Frankfurt (Macromolecular Complexes; ERG), the Collaborative Research Center 807 (Transport and Communication across Biological Membranes; GE2841/1-1; ERG), and FOR5046 (Integrated analysis of epithelial SLC26 anion transporters; GE2841/3-1; ERG), and the Swiss National Science Foundation (310030_188817; MAS). All members of the Geertsma and Pos laboratories are acknowledged for help in all stages of the project.

## Author contributions

B.T.K. and E.R.G. conceived the project. D.H.Ö., T.G., and B.T.K. generated the expression vectors for all uraA mutants. D.H.Ö., T.G., and B.T.K. purified wild type UraA, and the UraA mutants G112P, G320P, and P330G. D.H.Ö., T.G., and B.T.K. performed transport assays. T.G., D.H.Ö., and B.T.K. performed D.S.F. measurements. B.T.K., E.R.G., and I.Z. conducted sybody selections. I.Z. and M.A.S. supervised sybody selections. B.T.K. and E.R.G. purified sybodies. B.T.K. crystallized the Sy45-UraA(G320P) and Sy45-UraA(WT) complexes, collected diffraction data, processed the data and built and refined the models of Sy45-UraA(G320P) and Sy45-UraA(G320P)-Uracil. B.T.K. and E.R.G. analyzed the structures. B.T.K. performed the single cysteine accessibility assay. B.T.K. performed the scintillation proximity assay. J.Z. performed the HDX-MS on protein samples prepared by B.T.K. J.D.L. supervised the HDX-MS. B.T.K. and E.R.G. wrote the first draft of the manuscript. J.Z. and J.D.L. wrote the HDX-MS section and prepared the corresponding figures. All authors participated in the drafting and revision of the manuscript.

## Funding

## Competing interests

The authors declare no competing interests.
