## [Peer Review File · Nature Communications]

Interdomain-linkers control conformational transitions in the SLC23 elevator transporter UraAREVIEWER COMMENTS

Reviewer #1 (Remarks to the Author):

The manuscript by Kuhn and colleagues explores the regulatory role of interdomain linkers in the SLC23 elevator transporter UraA.

One of my main concerns is the impact of the differences in quaternary structures of the mutants vs. the WT UraA and the effect that these differences might have on the cell-based and in vitro studies.

Below detailed comments and questions:

-p.7 first sentence of the par. related to validating the conformational probe "As the tertiary and quaternary structure of UraA(G320P) and UraA(P330G) appeared intact ..." I am not sure how the quaternary structure is intact: if you compare the SEC profiles of both mutants vs. WT UraA you can clearly see that it is not. Can the authors please comment?

- Stoichiometry determined by SEC:

- i) How did the authors determine the stoichiometry from the SEC data? They only have elution volumes for a membrane protein in detergent micelles ...
- ii) From the SEC profiles, purified WT and G112P mutant are mainly eluting at around 13mL and there are nearly no proteins eluting at around 12mL. I am therefore not quite sure how the authors can state in p.6 "Decylmaltoside-solubilized wild type UraA eluted in two peaks with the monomeric species being most abundant [Fig. 1E]." The "dimeric" form is nearly non-existent.

- How can the authors exclude that stabilising/constraining a specific oligomeric state - clearly obvious for the mutants by SEC - does not have a negative impact on the transport measured in vivo? therefore, the impact of the tested mutations on the function could come from the disruption of the normal equilibrium between monomeric and dimeric protein rather than the impact on the intrinsic conformation of the analysed transporter.

- Specificity of Sy45: can the authors please comment on the lack of stabilisation of Sy45 of the P330G mutant compared to the effect with the G320P mutant (figure S6)? Why doesn't Sy45 bind to P330G mutant?

-Structure with Sy45:

- i) How can the authors rule out the fact that the WIO conformation might be induced by Sy45 binding?
- ii) A structure with WT UraA/Sy45 would've been nice to validate the WIO conformation with the WT protein. The WT/Sy45 did crystallise, couldn't the authors go forward with the structure refinement?

- Fig. S11: can the authors comment on the loss of nearly 50% of transport capacity of the L34C and V248C mutants? How would that impact the labelling and the sampled conformations by mPEG5K?

- Can the authors please specify if they did the binding experiments and the HDX experiments with the "dimeric" or the "monomeric" form - this is not clearly specified and therefore can impact all the data interpretation, especially that the ratio between both forms is not the same in the WT, G320P and P330G protein (if they did their analysis with a mixture of both forms).

- Differential HDX analysis: again, the form of the analysed protein (which SEC-resulting peak used) plays a crucial role in the HDX results. This should be clearly mentioned in the text.

Other comments on the HDX data below:

- i) Adjust scale bar to the maximum differential HDX observed for each analysed protein in figure 5: three different proteins and no back-exchange corrections result in qualitative data; this data enables

the comparison of a single protein in two states. No possible comparison between the 3 proteins when it comes to the absolute % differences.

ii) Is the effect on the upper part of TM1 (reduced deuterium uptake) of G320P statistically significant?

iii) The authors state at the end of their results: "We refrain from detailed analysis due to the unclear consequences on HDX of the increased dimer stability observed for these mutants [Fig. 1E]." the increased stability of the mutations will impact the overall dynamics of the protein, and not only in the scaffold domains.

iv) the authors should specify the concentration of protein used, the sequence coverage obtained after data curation, and the different filters used in DNX upon selecting the peptide list. Also, it would be nice to follow the recommendations of the HDX-MS community (Nature methods 2019, 16(7):595) for the supplementary material.

v) In general, it is a good idea to perform at least 2 biological replicates to rule out non-reproducible deuteration differences.

- The authors state in the discussion: "Consequently, it seems reasonable to assume that not the conformational space itself, but its occupancy is altered for UraA(G320P) [Fig. 6A], resulting in a reduced frequency of reaching one of the states needed to complete the transport cycle and thereby preventing reliable detection of transport." Can the authors comment on the impact of changing the monomer/dimer equilibrium on their conclusion?

Reviewer #2 (Remarks to the Author):

This is an interesting study by Kuhn et al into how residues in the linkers between two domains of the transporter, UraA influence the "elevator" mechanism of the protein. While the study is specific to the uracil transporter from the SLC23 class, given that other classes of transporter share the same topology and others still, the "elevator mechanism", their results are applicable to a wide range of transporters. They use a combination of site-directed mutagenesis, crystallography, binding and transport assays, alkylation and hydrogen deuterium exchange to provide a solid body of evidence that residues in the linkers can influence transport. In general, the paper is written and illustrated well.

Major comments:

They state that they have solved the structure of the G320P mutant in combination with the nanobody with and without uracil. This is at a relatively modest resolution of 3.5-3.7Å. I would like to see some evidence of the quality of the density around the binding site for the two maps showing one being apo and the other uracil-bound. At this resolution it is easy to misinterpret the density.

There appears to be a correlation between the stability of the protein and the propensity to form dimers. The mutants change the amount of dimer formed, with the mutants pushing the equilibrium towards the dimeric form. However, the crystal structures are monomeric. This may be due to the extra NG they add during crystallisation, but may also be due to the nanobody. I don't see that they have shown whether the nanobody affects the monomer/dimer equilibrium. This could affect the interpretation of the results in figure 4.

They show that the mutations affect transport in cells, but that various conformational states can be sampled in detergent solution. It is possible that the protein can take the conformational states in detergent but that this is more difficult within the confines of the lipid membrane. This should be discussed.

Minor comments

The CC0.5 of the two structures is ~ 1 in the highest resolution bin and the $I/\sigma I$ is quite high? If this is not due to the data collection strategy, could the resolution be pushed further?

Bottom of P8: the rmsds are shown for the two domains separately, but the rmsd for the full protein is only shown when compared to UapA and here it is relatively high. As worded and illustrated, there is no evidence to back up the claim that the conformation that they observe is most similar to that of UapA.

Last paragraph P13: some improvement in the wording may help the reader.

Bottom of P15: is the discussion with MBP relevant? In MBP the binding site is located between the two domains. In UraA it is in one domain.

REVIEWER COMMENTS

Reviewer #1 (Remarks to the Author):

The manuscript by Kuhn and colleagues explores the regulatory role of interdomain linkers in the SLC23 elevator transporter UraA.

One of my main concerns is the impact of the differences in quaternary structures of the mutants vs. the WT UraA and the effect that these differences might have on the cell-based and in vitro studies.

This is a valid concern that we would like to address immediately.

Regarding the cell-based assays: the quaternary state of UraA in the membrane is known to affect the transport function: wildtype UraA embedded in the membrane forms mostly, if not exclusively, dimers; mutants that monomerize UraA (as shown in detergent) no longer transport substrate (Fig. 4f and 5b in Yu et al., 2017). For this reason, we refrain from a more detailed analysis of the G112P mutant, which showed an increased monomer fraction (in detergent). Furthermore, Yu et al. (2017) also show that obligate dimers (two protomers fused by a short linker) transport substrate with an activity that appears to slightly exceed that of wild type UraA (note: in these experiments UraA is quantified by immunoblotting, which introduces some uncertainty as to whether the specific activity is truly increased or not). Thus, the fact that the G320P and P330G mutants show an increased fraction of dimers (in detergent) alone cannot explain their inability to transport uracil in cell-based studies.

We have updated the text on page 5 (line 123) to make this clear:

“Compared to wild type UraA, the fraction of dimers was reduced for UraA(G112P). Given that monomeric mutants of UraA are transport-incompetent¹³, this may underlie the lack of transport activity observed for this mutant. In contrast, the dimer content was greatly increased in UraA(G320P) and dimers were the dominant species in UraA(P330G). Since constitutive UraA dimers have high transport activities¹³, it is less obvious why these mutants are inactive.”

Regarding the impact of differences in the quaternary state of the mutants on in vitro studies: We have essentially used two types of in vitro studies, those related to substrate binding and those related to conformational dynamics. Yu et al. (2017) determined the uracil binding parameters of (detergent-solubilized) wild type UraA and the monomeric mutants mentioned above, using the same assay (SPA) as we used (Fig. 5a in Yu et al., 2017). They conclude that ‘[...] monomeric mutants of UraA bind to uracil with similar affinities as WT protein [...]’. Since their wild type UraA shows even higher fractions of dimeric protein, we conclude that the effect of the oligomeric state on the uracil binding affinity is negligible.

To make this clear we have updated the text on page 5 (line 132) and on page 8 (line 220):

“A similar degree of ligand-induced thermostabilization was observed for the G112P, G320P, and P330G mutants, indicating that these variants are still capable of binding substrate. Furthermore, these data are consistent with previous observations that the oligomeric state itself does not affect uracil binding properties¹³.”

“The notion that UraA(G320P) populates a different section of the transporter’s conformational space is further supported by its four-fold higher uracil binding affinity [Fig. 4D]. Again, addition of Sy45 reverted the variant back to a state with an affinity very similar to wild type UraA¹³. Notably, it has previously been shown that constitutive monomeric UraA mutants have binding affinities similar to the wild type protein¹³, making it unlikely that these changes in affinity are the result of an altered oligomeric state.”

Regarding the effect of differences in the quaternary state on our cysteine accessibility and HDX experiments: Given the extent of the dimer interface, which involves a full face of the scaffold domain, the conformational dynamics in the scaffold domain may be somewhat affected by increased dimerization. Nevertheless, the low RMSD of 1.9 Å for the comparison of the scaffold domains in monomeric UraA_{W10} and dimeric UraA_{OCC} suggests that the structure is not much affected by the

quaternary state. However, to exclude any influence of the oligomeric state on the conformational dynamics of the scaffold domain, we refrained from (over)interpreting the HDX data of the scaffold domain and placed our cysteine mutations in the transport domain.

To make this clear we have updated the text on page 7 (line 202) and page 10 (line 257):

“These positions were selected based on their accessibility either in our wide inward-open UraA structure or in an outward-open model³⁹ built using SLC4A1 as template [Fig. S12]¹⁰. Leu-34 and Val-248 are not part of the Sy45 epitope nor surface-exposed in the occluded conformation. We chose positions in the transport domain rather than the scaffold domain to minimize any impact of changes in the oligomeric state on the conformational dynamics.”

‘Although changes in deuterium uptake in the scaffold domain helices TM5 and TM12 appear to be consistent with altered conformational space occupancy in UraA(G320P) and UraA(P330G), we refrain from a detailed analysis due to the unclear consequences on the solvent accessibility of the different monomer-dimer equilibrium observed for these mutants [Fig. 1E].’

Finally, it could be argued that the oligomeric state could potentially affect the conformation of the transport domain relative to the scaffold domain. There are several problems with this assumption. First, any strong conformational preference of protomers in the dimer would render the wild type transporter inactive in the membrane where the dimer is the predominant state. Second, we observe only two discrete oligomeric states in detergent (the monomer and the dimer). However, our cysteine accessibility studies suggest at least three conformations (inward-open, outward-open, and occluded). Assuming that detergent-solubilized dimeric UraA(P330G) exclusively adopts the occluded conformation and monomeric UraA (wildtype) adopts an inward-open conformation, one would expect the G320P mutant (i.e., more dimer than WT UraA and more monomer than P330G) to adopt a conformation in between. Instead, it adopts the outward-open conformation.

Taken together, this strongly suggests that the strongest effect of the mutations lies in their ability to control the conformational landscape of the transporter. The observed changes in the oligomeric state appear to be an interesting consequence of the altered conformational landscape, but not its cause, and the functional outcome largely depends on the conformational changes. To communicate this more clearly, we have updated the text on page 12 (line 318; Discussion):

“UraA(G320P) in complex with Sy45 crystallized in a novel wide inward-open structure. The high similarity of the scaffold domain in UraA_{WIO} and UraA_{OCC} suggests that the structural differences with UraA_{IO}, in particular the large apparent rearrangement of TM6 and TM7, are not part of the linear transport mechanism as suspected previously¹³. However, the scaffold domain does not appear to be as rigid as the transport domain. We observe a small shift of the cytoplasmic side of TM6-TM7 to the periphery of the dimer that provides space for TM1 in the transport domain to close the inside cavity during the UraA_{WIO}-UraA_{OCC} transition. Similar subtle adaptations of the scaffold domain in response to mutation-induced changes in the conformational landscape of the transport domain may contribute to the enhanced stability of the UraA(G320P) and UraA(P330G) dimers. Furthermore, the increased dimer stability among our mutants with decreased conformational flexibility suggests some degree of conformational ‘breathing’ in the scaffold domain during transport, in line with the reduced transport rates observed upon cross-linking the scaffold domains of the SLC13 transporter VcINDY⁴⁴.”

Below detailed comments and questions:

-p.7 first sentence of the par. related to validating the conformational probe “As the tertiary and quaternary structure of UraA(G320P) and UraA(P330G) appeared intact ...” I am not sure how the quaternary structure is intact: if you compare the SEC profiles of both mutants vs. WT UraA you can clearly see that it is not. Can the authors please comment?

We agree with the reviewer that the oligomeric state of the G320P and P330G mutants differs from that of the wild-type protein: both mutants appear to form stronger dimers. Our use of the term “intact” refers to our concern that the respective mutations would have monomerized the protein, an

oligomeric state in which UraA has previously been shown to be inactive (Yu et al., 2017). Clearly, monomerization is not the cause of the lack of transport activity. We have rephrased this section accordingly (page 5, line 140).

“As the loss of function in UraA(G320P) and UraA(P330G) cannot be attributed to monomerization [Fig. 1E] or impaired substrate binding [Fig. 1F], we focused on uncovering potential anomalies in the conformational dynamics of these UraA mutants.”

- Stoichiometry determined by SEC:

i) How did the authors determine the stoichiometry from the SEC data? They only have elution volumes for a membrane protein in detergent micelles ...

We did not determine stoichiometry from the SEC data, but rather determined the fractions of detergent-solubilized protein present in dimeric and monomeric states during SEC. The assumption that these two peaks correspond to monomeric and dimeric decylmaltoside-solubilized UraA is based on static light scattering studies from the Nieng Yan lab (Yu et al., 2017).

ii) From the SEC profiles, purified WT and G112P mutant are mainly eluting at around 13mL and there are nearly no proteins eluting at around 12mL. I am therefore not quite sure how the authors can state in p.6 “Decylmaltoside-solubilized wild type UraA eluted in two peaks with the monomeric species being most abundant [Fig. 1E].” The “dimeric” form is nearly non-existent.

We agree that the dimeric form of detergent-solubilized wildtype UraA is not well-visible in the graph as a result of the small size and the comparably thick lining. However, we reproducibly observe a small ‘hump’ preceding the monomeric species upon SEC. Given the high purity of the sample and the substantial separation of the peaks, we were able to deconvolute these allowing us to estimate the fraction in each oligomeric state. We have added this data as an additional supplementary table [Table S1] (page 38), which is referred to on page 5 (line 122).

‘Decylmaltoside-solubilized wild type UraA eluted in two peaks with the monomeric species being most abundant [Fig. 1E, Table S1].’

Table S1: Monomer-dimer ratio of decylmaltoside-solubilized UraA variants in size exclusion chromatography. The fraction of protomers migrating as monomers and dimers was calculated following peak deconvolution using Origin.

UraA variant	Monomer (%)	Dimer (%)
WT	95	5
G112P	97	3
G320P	75	25
P330G	18	82

- How can the authors exclude that stabilising/constraining a specific oligomeric state - clearly obvious for the mutants by SEC - does not have a negative impact on the transport measured in vivo? therefore, the impact of the tested mutations on the function could come from the disruption of the normal equilibrium between monomeric and dimeric protein rather than the impact on the intrinsic conformation of the analysed transporter.

We have addressed this point above:

Regarding the cell-based assays: the quaternary state of UraA in the membrane is known to affect the transport function: wildtype UraA embedded in the membrane forms mostly, if not exclusively, dimers; mutants that monomerize UraA (as shown in detergent) no longer transport substrate (Fig. 4f and 5b in Yu et al., 2017). For this reason, we refrain from a more detailed analysis of the G112P mutant, which showed an increased monomer fraction (in detergent). Furthermore, Yu et al. (2017) also show that obligate dimers (two protomers fused by a short linker) transport substrate with an activity that appears to slightly exceed that of wild type UraA (note: in these experiments UraA is quantified by immunoblotting, which introduces some uncertainty as to whether the specific activity is truly

increased or not). Thus, the fact that the G320P and P330G mutants show an increased fraction of dimers (in detergent) alone cannot explain their inability to transport uracil in cell-based studies.

We have updated the text on page 5 (line 123) to make this clear:

'Compared to wild type UraA, the fraction of dimers was reduced for UraA(G112P). Given that monomeric mutants of UraA are transport-incompetent¹³, this may underlie the lack of transport activity observed for this mutant. In contrast, the dimer content was greatly increased in UraA(G320P) and dimers were the dominant species in UraA(P330G). Since constitutive UraA dimers have high transport activities¹³, it is less obvious why these mutants are inactive.'

In addition, the reviewer may have considered the observations from Mulligan et al. (2015) on VcINDY mutants that are covalently crosslinked across their scaffold domain. These 'stapled' mutants also obligatorily form strong dimers. However, although this stapling resulted in some reduction in the transport rate, it did not abolish transport as we observe here for UraA-G320P and -P330G.

Taken together, we conclude that the impact of the respective mutations lies in its effect on the conformational space of the transporter, and that the observed changes in the oligomeric state are a consequence of the altered conformational landscape, but not its cause.

- Specificity of Sy45: can the authors please comment on the lack of stabilisation of Sy45 of the P330G mutant compared to the effect with the G320P mutant (figure S6)? Why doesn't Sy45 bind to P330G mutant?

As indicated in the main text (page 8, line 226), we indeed attribute the lack of P330G stabilization to a reduced binding efficiency of Sy45.

"The addition of Sy45 did not significantly affect the conformational equilibrium [Fig. 4B], nor increased the melting temperature of the mutant substantially [Fig. 4C, Fig. S6], suggesting a strongly reduced binding efficiency of Sy45 to UraA(P330G)."

We expect that this reduced binding efficiency is due to the fact that UraA(P330G) spends most of its time in a conformation that is close or identical to the occluded conformation, as suggested by our cysteine accessibility studies and our HDX MS experiments. However, since uracil can still bind, this mutant must spend at least a fraction of its time in one 'open' conformation (either inward-, or outward-open). We expect that on the time scale of our experiments (minutes), the inward-open conformation is not sampled sufficiently to allow Sy45 binding and thermostabilization of a significant fraction of the protein population.

Occasionally it is suggested that binders can force a protein into a particular conformation, and it could be argued that the lack of Sy45 binding to UraA(P330G) is surprising in this light. However, we consider such binder-induced conformational changes to be highly unlikely in general. Forcing a protein to change conformation requires energy. Since this energy can only come from the binding event itself, binders with this ability are expected to have extremely low affinity and are unlikely to bind under standard conditions. Instead, we expect binders to stabilize a conformation that the protein spontaneously adopts, thereby shifting the equilibrium towards that particular state. We interpret the lack of Sy45-induced stabilization of P330G as a strong indication that this mutant rarely, if ever, occupies the WIO state.

-Structure with Sy45:

i) How can the authors rule out the fact that the WIO conformation might be induced by Sy45 binding?

See above. Due to the energetic penalty of forcing a protein into a particular state, such binders are expected to have particularly poor affinities. Rather, binders stabilize conformations that the protein spontaneously visits, thereby shifting the conformational equilibrium toward such a conformation.

Furthermore, given that Sy45 was the outcome of a selection against wild type UraA, we expect this binder to recognize a native conformation.

ii) A structure with WT UraA/Sy45 would've been nice to validate the WIO conformation with the WT protein. The WT/Sy45 did crystallise, couldn't the authors go forward with the structure refinement?

We agree with the reviewer that a structure with UraA(WT)/Sy45 would have further validated the WIO conformation. However, in contrast to single particle cryoEM, the efficiency of sample optimization for X-ray crystallography is a particularly disappointing and slow process. We were unable to optimize our UraA(WT) crystals to a resolution sufficient for structure refinement.

However, we are confident that our structure represents a native conformation. First, Sy45 was selected using wild type UraA and is therefore expected to stabilize a conformation part of the conformational space of wild type UraA. Second, given that the rigid body elevator mechanism is based on relative reorientations of the transport and scaffold domains, we believe that the low RMSD of these individual domains with respect to the published occluded structure is sufficiently strong evidence for a good and representative structure.

- Fig. S11: can the authors comment on the loss of nearly 50% of transport capacity of the L34C and V248C mutants? How would that impact the labelling and the sampled conformations by mPEG5K?

This figure is now labeled Fig. S13 (page 31).

We have not extensively investigated the underlying cause of this difference in transport activity, as this particular experiment was designed to determine whether or not the respective mutants are transport competent. It is clear that all mutants are active. Therefore, it is reasonable to assume that all relevant conformations required for transport are sampled by these mutants, justifying their use.

Importantly, we determined the accessibility of these substituted cysteines in three different contexts (wildtype, G320P, and P330G). This means that any unwanted effect should be compensated by our relative measurement. Furthermore, we used two different positions for which opposite effects were expected. For example, the increased accessibility of Cys-34 and the reciprocal decreased accessibility of Cys-248 observed for G320P compared to wild type can only be interpreted as a shift from the more inward-open conformation (for wildtype) to a more outward-open conformation for the G320P mutant.

Finally, we independently validated our conformational space claims using the complementary approach of HDX MS, which does not require any additional mutations or labels.

Concerning a potential cause for the reduced transport activity: the respective mutants are generated in the background of a cysteine-less derivative of UraA. This mutant, designated UraA^{CL}, contains four mutations: C61S, C97S, C102S, and C318A. While we have not explicitly studied the effect of these mutations alone or in combination, we anticipate that the number and nature of these mutations may have reduced the overall transport rate relative to the wild type transporter. We have now updated the legend to include the additional mutations required to generate UraA^{CL}. We have also included an additional panel demonstrating that the expression levels of the respective mutants are not strongly affected.

Fig. S13: (A) Transport rates of [³H]-uracil by UraA variants in *E. coli* BW25113(Δ uraA) with three technical replicates shown as scatter plot and derived mean values \pm SER as bars. **(B)** In-gel fluorescence of SDS-PAGE gel loaded with the samples derived from the cells used for the transport assay. Single cysteine mutants L34C, L85C, and V248C were generated in a cysteine-less background (UraA(C61S/C97S/C102S/C318A), hereafter referred to as UraA^{CL}). All UraA derivatives (1: UraA(WT); 2: control UraA(E241A/H245A/E290A); 3: UraA^{CL}(L34C); 4: UraA^{CL}(L85C); 5: UraA^{CL}(V248C)) were expressed from the pBXC3GH plasmid as a C-terminal GFP fusion protein.

- Can the authors please specify if they did the binding experiments and the HDX experiments with the “dimeric” or the “monomeric” form – this is not clearly specified and therefore can impact all the data interpretation, especially that the ratio between both forms is not the same in the WT, G320P and P330G protein (if they did their analysis with a mixture of both forms).

We thank the reviewer for pointing this out. Although we do not expect the oligomeric state to play a causal role in the function and conformational dynamics as eluted above, we should have clarified this.

For all experiments with detergent-solubilized derivatives of UraA, we used the major SEC peak, e.g. the fractions representing monomers for UraA(WT) and UraA(G112P), and the fraction representing dimers for UraA(P330G). In the case of UraA(G320P), the fractions representing the dimers and the monomers were pooled. These pooled fractions were then concentrated as indicated in the Materials and Methods. We have specified this in the Materials and Methods section (page 14, line 385).

The concentrated sample was centrifuged for 10 min at 13,000 x g to remove large aggregates and subsequently loaded on a Superdex 200 increase 10/300 GL size exclusion column equilibrated with purification-buffer supplemented with 0.2% (w/v) DM or 10 mM Hepes pH 7.5, 150 mM NaCl, 0.2% DM (w/v) in case the sample was used for crystallization. Peak fractions (UraA(WT) and UraA(G112P): monomeric fraction; UraA(P330G): dimeric fraction; UraA(G320P): monomeric and dimeric fractions) were pooled and concentrated.

Concerning the impact of the oligomeric state on the uracil binding, the lab of Nieng Yan (Yu et al., 2017) has already demonstrated that the oligomeric state of UraA does not affect its uracil binding constant.

Concerning the impact of the oligomeric state on the HDX experiments, please see our reply to point iii) below.

We would like to point out that detergent-solubilized membrane proteins are expected to undergo reversible self-association with a defined stoichiometry (see e.g. <https://doi.org/10.1074/jbc.m004066200>). For UraA, this means that the fraction of monomers and dimers depends on the protein concentration, which in turn depends on the experimental conditions (HDX: 0.045 mg/ml; cysteine alkylation: 0.6 mg/ml; DSF: 0.1 mg/ml; SPA: very high (>>5 mg/ml) given that the protein is concentrated on a surface). The association constant for UraA in 0.2% decylmaltoside is not known and, based on our data, the G320P and P330G mutants are expected to have different association constants. Regrettably, the determination of association constants for membrane proteins is not trivial and represents a significant amount of work. Taken together, this means that we cannot be certain about the proportion of monomers and dimers in our experiments. However, as mentioned above, we have taken this into account in the design of our experiments (e.g. no cysteines in the scaffold domain) and in the evaluation of the experimental results (we specifically refrained from a detailed interpretation of the HDX data on the scaffold domain).

- Differential HDX analysis: again, the form of the analysed protein (which SEC-resulting peak used) plays a crucial role in the HDX results. This should be clearly mentioned in the text.

We completely agree and thank the reviewer for pointing this out. As mentioned above, we have included this information in the revised manuscript (page 14, line 385).

Other comments on the HDX data below:

i) Adjust scale bar to the maximum differential HDX observed for each analysed protein in figure 5: three different proteins and no back-exchange corrections result in qualitative data; this data enables the comparison of a single protein in two states. No possible comparison between the 3 proteins when it comes to the absolute % differences.

We thank the reviewer for raising this valid question. However, we used the same system with identical parameters for all experiments, and conducted all experiments within a few days of each other. We do not make claims about absolute deuterium uptake rates or solvent accessibility, which is impossible

without back correction, as the reviewer correctly pointed out. We do compare relative uptake rates which are affected identically by back exchange in our experiments. We thus suggest to keep the figure “as-is”, but prepared an additional figure with maximum differential scale for reference [Fig. S17] (page 37) that is referred to on page 9, line 242.

‘For wild type UraA, the addition of 100 μ M uracil induced a decrease in deuterium uptake in the substrate binding site (TM3 and TM10) and adjacent regions (TM1 and TM8) [Fig. 5A, Fig. S17].’

Fig. S17: Hydrogen-deuterium exchange mass spectrometry analysis of UraA variants with scale bar adjusted to individual datasets. (A) HDX of wild type UraA in the presence and absence of 100 μ M uracil. Transport and scaffold domain are oriented as if the protein was folded open as indicated in the central panel. The interdomain-linkers are shown in both domains as a reference point. The circle on the scaffold domain represents the position of the substrate in the opposing transport domain assuming the UraA_{occ} conformation. **(B)** Substrate-dependent HDX for UraA(G320P) and UraA(P330G). **(C)** Differential HDX resulting from the G320P (top panel) and P330G (lower panel) mutations in the absence of ligand compared to wild type UraA. The color range representing the relative deuterium uptake, ranging from red (decreased uptake) over white (no change) to blue (increased deuterium uptake), applies to all panels. Grey stretches indicate regions of not identified amino acids.

ii) Is the effect on the upper part of TM1 (reduced deuterium uptake) of G320P statistically significant?

Yes, every differential deuterium uptake of every peptide was subjected to a two staged T-test (first stage $p=0.05$ and second stage $p=0.01$). Only peptides passing both stages were classified as statistically significant and plotted onto the structure.

iii) The authors state at the end of their results: “We refrain from detailed analysis due to the unclear consequences on HDX of the increased dimer stability observed for these mutants [Fig. 1E].” the increased stability of the mutations will impact the overall dynamics of the protein, and not only in the scaffold domains.

We thank the reviewer for raising this issue. This is a question of causality. The relationship between oligomer stability and conformational dynamics is not established. Considering that 1: the transport domain is not part of the dimer interface and can in principle move freely; 2: the scaffold domain shows only minor rearrangements during the conformational transition; and 3: the positions of the introduced mutations in the UraA structure are not located within the dimerization interface, we think it is more likely that the conformational dynamics of the transporter affect the oligomer stability in detergent and not vice versa. However, we cannot exclude that the solvent accessibility of the side of the scaffold domain facing the dimer interface is slightly different in monomer and dimer. Therefore, we refrain from a detailed analysis. We have reformulated this section accordingly (page 10, line 257).

‘Although changes in deuterium uptake in the scaffold domain helices TM5 and TM12 appear to be consistent with altered conformational space occupancy in UraA(G320P) and UraA(P330G), we refrain from a detailed analysis due to the unclear consequences on the solvent accessibility of the different monomer-dimer equilibrium observed for these mutants [Fig. 1E].’

iv) the authors should specify the concentration of protein used, the sequence coverage obtained after data curation, and the different filters used in DNX upon selecting the peptide list. Also, it would be nice to follow the recommendations of the HDX-MS community (Nature methods 2019, 16(7):595) for the supplementary material.

For all HDX MS experiments we used a starting protein concentration of 0.675 mg/mL which equals to 15 μ M. At the start of the experiment, 4 μ l of this sample was diluted in 56 μ l L- or E-buffer yielding a protein concentration of 0.045 mg/ml. This information was provided in the “HDX sample preparation” section on page 16, but we have clarified it in the revised manuscript (page 16, line 488)

“For HDX-MS experiment a fully automated HDX-2 system (Waters, Milford USA) was used. After equilibration of the sample and all the buffers, the experiment was started by diluting 4 μ L of sample in 56 μ L (resulting in a protein concentration of 0.045 mg/ml) of either E- buffer, for non-deuterated reference measurements, or in L buffer for five time points (0, 30, 360, 900 and 2700 seconds of labelling).”

The maps shown in supplementary Fig. S15 already represent the sequence coverage after data evaluation with DynamX. We have now clarified this in the figure legend (page 33):

“Fig. S15: Sequence coverage maps of all UraA variants with its corresponding peptide numbers and redundancy. Data shown is the sequence coverage after data evaluation with DynamX. Peptides are shown in green bars. No statistical tests were applied to generate the coverage maps.”

We now provide additional tables for all experiments, formatted according to the recommended paper. These tables are provided as supplementary files [Table S4.-Table S8].

These tables are referred to on page 17, line 531.

‘Supplementary Tables S4-S8 provide HDX data tables formatted according to the recommendations of the HDX-MS community⁴⁰.’

⁴⁰ Masson et al., 2019

We provide the DynamX filter settings in the new supplementary table S3 (below).

This table is referred to on page 16, line 511.

‘For peak picking of all corresponding peaks through calculating weighted average m/z (centroid) for all isotopic peaks in each peptide DynamX 3.0. (Waters) was used [Table S3].’

Table S3: Filter setting in DynamX

Parameter	Value
Minimum intensity	0
Minimum sequence length	3
Maximum sequence length	20
Minimum products	2
Minimum products per amino acid	0.1
Minimum consecutive products	0
Minimum sum intensity for products	0
Minimum score	6
Maximum MH+ Error (ppm)	0
File threshold	6
Retention time RSD	0
Intensity RSD	0

v) In general, it is a good idea to perform at least 2 biological replicates to rule out non-reproducible deuteration differences.

The reviewer is correct. However, the extensive nature of the data generation and analysis for these three samples prevented us from performing two biological replicates. For our experimental HDX-MS setup, we have acquired data from 4 technical replicates, demonstrating the reproducibility of our method (with average RSDs below 3% for all experiments). For our biological sample preparation, we took great care to ensure that the protein samples were in excellent condition. Evidence that the proteins are indeed in a well-folded state at the time of analysis is provided by the fact that the overall HDX pattern is consistent with the 3D structure. We performed the binding experiments under conditions where the protein was shown to be active (as shown by the uracil-induced T_m shift in DSF and the uracil-binding observed in SPA), and observed that substrate addition induced a change in HDX consistent with the expected function of the protein. Finally, the changes in HDX mirror the changes we observed using the cysteine accessibility assay. Therefore, we consider the data to be of sufficient quality. Should future developments in instrumentation and data analysis allow the inclusion of biological replicates, we will strongly consider this as suggested by the reviewer.

- The authors state in the discussion: “Consequently, it seems reasonable to assume that not the conformational space itself, but its occupancy is altered for UraA(G320P) [Fig. 6A], resulting in a reduced frequency of reaching one of the states needed to complete the transport cycle and thereby preventing reliable detection of transport.” Can the authors comment on the impact of changing the monomer/dimer equilibrium on their conclusion?

We do not expect any impact and refer to our extensive discussion on the impact of the quaternary state above.

The question of the reviewer is one of causality. Does the oligomeric state affect the conformational dynamics or do the conformational dynamics affect the oligomeric state? The relationship between oligomer stability and conformational dynamics is not established. Considering that 1: the transport domain is not part of the dimer interface and can in principle move freely; 2: the scaffold domain shows only minor rearrangements during the conformational transition; and 3: the positions of the introduced mutations in the UraA structure are not located within the dimerization interface, we think it is more likely that the conformational dynamics of the transporter affect the oligomer stability in detergent and not vice versa. We feel supported by the observations from the Nieng Yan lab (Yu et al., 2017) that demonstrated identical uracil binding affinities for monomeric mutants and the wild type protein. Any significant change in the conformational space of these monomeric mutants would have been expected to affect substrate binding as well.

Reviewer #2 (Remarks to the Author):

This is an interesting study by Kuhn et al into how residues in the linkers between two domains of the transporter, UraA influence the “elevator” mechanism of the protein. While the study is specific to the uracil transporter from the SLC23 class, given that other classes of transporter share the same topology and others still, the “elevator mechanism”, their results are applicable to a wide range of transporters. They use a combination of site-directed mutagenesis, crystallography, binding and transport assays, alkylation and hydrogen deuterium exchange to provide a solid body of evidence that residues in the linkers can influence transport. In general, the paper is written and illustrated well.

We kindly thank the reviewer for acknowledging the broader implications of our study for other elevator transporters. We appreciate the reviewer’s acknowledgement that our use of complementary approaches has resulted in a solid body of evidence that strengthens our claim for the relevance of interdomain linkers.

Major comments:

They state that they have solved the structure of the G320P mutant in combination with the nanobody with and without uracil. This is at a relatively modest resolution of 3.5-3.7Å. I would like to see some evidence of the quality of the density around the binding site for the two maps showing one being apo and the other uracil-bound. At this resolution it is easy to misinterpret the density.

Although the UraA(G320P)-Sy45 complex was co-crystallized in the absence of uracil, we do not consider this structure to be a true apo-state. This is due to the fact that we observe additional electron density in the substrate binding site of “apo”-UraA. We exclude that this is co-purified uracil, as we require the addition of uracil to observe thermostabilization [Fig. 1E]. We have modeled the buffer compound Tris in this density. The data are now provided as new supplementary figure [Fig. S8] (below).

This figure is referred to on page 6, line 161.

‘This is also reflected by a second crystal structure of UraA(G320P)-Sy45 in presence of uracil refined to 3.7 Å showing the transporter in the same inward-facing conformation [Fig S8].’

Fig. S8: Substrate binding site of UraA(G320P)-Sy45 apo (**A**) and uracil (**B**) structures. The 2Fo-Fc electron density maps shown as blue mesh are contoured at 1σ . Though the UraA(G320P)-Sy45 complex was co-crystallized in absence of uracil, additional electron density in the substrate binding site was observed. As substantial co-purification of uracil could be excluded based on the observed thermostabilization upon addition of uracil [Fig. 1E], we modeled the buffer compound Tris into the density.

Since, as pointed out by the reviewer, the electron density of Uracil and Tris is essentially indistinguishable at the given resolution of the collected X-ray diffraction data, we refrain from detailed analysis of the substrate binding site.

There appears to be a correlation between the stability of the protein and the propensity to form dimers. The mutants change the amount of dimer formed, with the mutants pushing the equilibrium towards the dimeric form. However, the crystal structures are monomeric. This may be due to the extra NG they add during crystallisation, but may also be due to the nanobody. I don't see that they have shown whether the nanobody affects the monomer/dimer equilibrium. This could affect the interpretation of the results in figure 4.

We agree with the Reviewer that there is an apparent discrepancy between the oligomeric states observed for UraA(G320P) in the SEC and in the structure. The monomer observed in the crystal structure results from a steric hindrance that prevents the conventional UraA dimer from binding two Sy45 protomers. The addition of Sy45 thus shifts the monomer/dimer equilibrium that exists in detergent toward the monomeric state.

Regarding the effect of Sy45-induced monomerization of detergent-solubilized UraA(G320P), we refer to our first comment to Reviewer #1 addressing the impact of differences in the quaternary state of the mutants on the interpretation of our results. In short, we do not expect these differences to affect our interpretation of the results in figure 4: Previously, the Nieng Yan lab did not observe any consequences of altered oligomeric states on substrate binding (Fig. 5a in Yu et al., 2017); Regarding the cysteine accessibility assay, positions L34C and V248C are located in the transport domain and are not involved in the dimer interface. We do not expect differences in the accessibility of these residues between the monomeric or dimeric state per se. The structural similarity between the UraA scaffold domain in the dimeric, occluded and monomeric, wide inward-open conformations further supports our assumption that the conformational dynamics within the scaffold domain are unaffected by the oligomeric state in the detergent-solubilized state. Finally, assuming that dimeric UraA exclusively adopts the occluded conformation and that monomeric UraA adopts an inward-open conformation, one would expect the G320P mutant (i.e., more dimer than WT UraA and more monomer than P330G) to adopt a conformation in between. Instead, it adopts the outward-open conformation.

We conclude that the strongest effect of the mutations lies in their ability to control the conformational landscape of the transporter. The observed changes in the oligomeric state appear to be an interesting consequence of the altered conformational landscape, but not its cause.

They show that the mutations affect transport in cells, but that various conformational states can be sampled in detergent solution. It is possible that the protein can take the conformational states in detergent but that this is more difficult within the confines of the lipid membrane. This should be discussed.

The reviewer raises an important point that applies to the entire field of membrane protein biochemistry: To what extent are our data obtained for detergent-solubilized proteins representative of these proteins embedded in a lipid bilayer? Ideally, we would characterize all relevant parameters of membrane proteins directly in representative lipid bilayers. However, as in our case, this is often not feasible due to experimental limitations.

The reviewer is correct assuming that we cannot exclude that UraA(G320P), which samples various conformations in detergent, may not sample the same conformations once it is embedded in a membrane.

To make this limitation explicit, we have modified the corresponding section in the discussion (page 11, line 282) to read:

'Within the limits of our experimental resolution, all transport-relevant conformations thus appear to be accessible. Assuming that our data on detergent-solubilized protein are representative of the behavior of the protein within the confines of the lipid membrane, it seems reasonable to propose that not the conformational space itself, but its occupancy is altered for UraA(G320P) [Fig. 6A]. This may lead to a

reduced frequency of reaching one of the states *required* to complete the transport cycle, thereby preventing reliable detection of transport.'

Minor comments

The CC0.5 of the two structures is ~1 in the highest resolution bin and the $I/\sigma I$ is quite high? If this is not due to the data collection strategy, could the resolution be pushed further?

We observed diffraction anisotropy and performed ellipsoidal truncation and anisotropic scaling as indicated in the Material and Methods section. The resolution could be pushed further but only at the expense of dataset completeness in the highest resolution shell.

Bottom of P8: the rmsds are shown for the two domains separately, but the rmsd for the full protein is only shown when compared to UapA and here it is relatively high. As worded and illustrated, there is no evidence to back up the claim that the conformation that they observe is most similar to that of UapA.

We thank the reviewer for pointing out this omission. We now provide an additional supplementary Fig. S10 illustrating that our UraA_{WIO} structure shows most resemblance to the published inward-open conformation of UapA (figure below).

This figure is referred to on page 6, line 169.

'With an RMSD of 4.1 Å over 358 residues, our structure resembles that of inward-facing UapA [Fig. S9C, Fig. S10]¹⁴. It displays a wider opening and consequently a larger cytoplasmic cavity (4780 Å³) than UraA_{IO} (3300 Å³)^{36,37}, hence we term this conformation wide inward-open (UraA_{WIO}) [Fig. S10, Fig. S11].'

Fig. S10: Cavity of UraA_{WIO}, UraA_{OCC}, UraA_{IO} and UapA shown as side view from within the membrane cut through in the plane of the substrate binding site (red asterisk) with transport and scaffold domain in green and purple color, respectively.

In addition, we have modified supplementary Fig. S9 to include the superimpositions of the full-length proteins as well (figure below)

Please note that the sequence identity between UraA and UapA is only 25.7%. UapA features long extracellular loops, a slightly different location of TM14 and an alpha-helical TM13-14 loop instead of the beta-sheet in UraA_{WIO} [Fig. S9]. This causes the high RMSD in our alignment of the full-length proteins. However, upon truncation of the TM2-3, TM3-4, TM5-6 and TM13-14 loops in UapA and the TM5-6 and TM13-14 loops in UraA, the similarity of the domain orientation between the two structures is more obvious and results in a lower RMSD of 4.1 Å (over 358 residues) instead of 6.5 Å over 397 residues. This is now mentioned on page 6, line 168.

'With an RMSD of 4.1 Å over 358 residues, our structure resembles that of inward-facing UapA [Fig. S9C, Fig. S10]¹⁴.'

Fig. S9: Structural alignments of SLC23 structures. Structural alignment of the transport (**A**) and scaffold domains (**B**) of relevant SLC23 structures using the default matchmaker command of UCSF Chimera⁶⁵. (**C**) Whole protein structural alignment. For the alignment with UapA we truncated the TM2-3, TM3-4, TM5-6 and TM13-14 loops in UapA and the TM5-6 and TM13-14 loops in UraA to emphasize the high similarity in the orientation of the α -helices. Without these truncations an RMSD of 6.5 Å over 397 residues is obtained.

Last paragraph P13: some improvement in the wording may help the reader.

We thank the reviewer for pointing this out. The respective paragraph now reads:

*“Changes in conformational dynamics are **best** visualized by directly comparing the relative deuterium uptake between the wild type protein and the mutants in their apo states [Fig. 5C]. UraA(G320P) showed an increase in HDX **on** the extracellular side of TM1 and a concomitant decrease **on** the cytoplasmic side of TM8. **These data are** in full agreement with our alkylation analysis [Fig. 4B]. While the differential deuterium uptake observed for UraA(P330G) **shows** similar agreement with our cysteine accessibility studies, the increased spatial resolution of HDX-MS highlights that deuterium uptake is **significantly** reduced in **almost the entire** side of the transport domain facing the scaffold domain [Fig. 5C]. In contrast to UraA(G320P), no increase in uptake **was observed** on the extracellular side of TM1, **consistent** with the assumption that this variant **adopts** an occluded conformation. **Although** changes in deuterium uptake in the scaffold domain helices TM5 and TM12 appear to **be consistent** with altered **conformational space occupancy** in UraA(G320P) and UraA(P330G), we refrain from a detailed analysis due to the unclear consequences **on the solvent accessibility of the different monomer-dimer equilibrium observed for these mutants [Fig. 1E].”***

Bottom of P15: is the discussion with MBP relevant? In MBP the binding site is located between the two domains. In UraA it is in one domain.

We believe that the comparison with the MBP hinge mutants is relevant, because this example clearly shows that affinity depends not only on the number and type of interactions between the substrate and the protein, but also on the ability of the substrate to leave the binding site following its unbinding. The hinge mutants in MBP decrease the opening frequency of the binding protein and thereby increase its affinity. Similarly, our interdomain linker mutants, in particular P330G, decrease the occupancy of the inward-open and outward-open states, thereby decreasing the efficiency with which the substrate leaves the binding site upon unbinding.

REVIEWER COMMENTS

Reviewer #1 (Remarks to the Author):

The authors did not convincingly address my main major concern and, in my opinion, additional experiments are required to consolidate their claim that "the observed changes in the oligomeric state appear to be an interesting consequence of the altered conformational landscape, but not its cause". Additional HDX-MS experiments with the non-pooled monomeric and dimeric G320P species separately as well as the main SEC fraction of the G112P can help validating their claim.

More specifically:

In p.5 line 123-125 the authors state "Compared to wild type UraA, the fraction of dimers was reduced for UraA(G112P). Given that monomeric mutants of UraA are transport-incompetent, this may underlie the lack of transport activity observed for this mutant."

I am not at all convinced with the quantification from one SEC profile of 5% and 3% of dimeric species for the WT and G112P proteins respectively in table S1. The authors are making conclusions based on a 2% difference without any replicates nor SDs... How did the author integrate these humpy profiles? Due to the very low amount of dimeric species (if present) and the limitation in SEC sensitivity, the authors should include at least a quantification of triplicate SEC data with corresponding SD values in table S1. I presume they repeated the expression/purification a sufficient number of times to include this data.

In the same line of concerns, the authors state in their rebuttal: "Taken together, we conclude that the impact of the respective mutations lies in its effect on the conformational space of the transporter, and that the observed changes in the oligomeric state are a consequence of the altered conformational landscape, but not its cause." The oligomeric state of the G112P versus WT is not significantly impacted judging from the SEC data; therefore, I am not convinced that an altered conformational landscape will lead to a change of the oligomeric state. To consolidate their statement, the authors should analyze the conformation of the G112P mutant (that does not lead to a significant change in the oligomeric state) to check if the conformation is altered or not. Therefore, HDX-MS analysis of G112P will be a highly valuable experiment to consolidate the authors conclusions.

In addition, SEC profile of G320P shows higher monomeric species compared to dimeric species; since the HDX-MS relate to the average of all states present in solution, the authors should repeat the experiments with the non-pooled monomeric G320P and dimeric G320P separately in order to demonstrate that the oligomeric state does not affect the overall HDX-MS data. Indeed, the differential HDX-MS profiles for apo versus uracil-bound proteins for WT and G320P mutant are very similar, and different compared to the P330G mutant (that gives mainly dimeric species); the WT VS. mutants comparison is also impacted by the fact that the analysis in the case of G320P was performed on the pooled monomeric and dimeric species.

Other minor comments:

-The authors should clearly state in the text that they refer to the study from the Nieng Yan lab (Yu et al., 2017) to assume the oligomeric state based on the SEC data (p.5, line 121/122).

- Fig. S16 is not really helpful to address the quality of the HDX-MS data; instead, the authors should consider adding overall uptake plots of the different peptides in compared states, which provides a better and complete representation.

Reviewer #2 (Remarks to the Author):

I have looked through the rebuttal and reread the paper. The authors have answered the major concerns.

The only further comment that I have regards the presence of uracil in the crystal structure. This does not really influence any of the major conclusions of the paper, but from the new supplementary Figure 8, it is not obvious that there is any difference between the "uracil bound" and the density of the Tris. There is 50mM Tris in both the crystal conditions. Is it possible that Tris is present in both cases? Density should really be shown from maps calculated before the addition of the ligands. If there is any uncertainty in interpretation this should be acknowledged in the text.

REVIEWER COMMENTS

Reviewer #1 (Remarks to the Author):

The authors did not convincingly address my main major concern and, in my opinion, additional experiments are required to consolidate their claim that “the observed changes in the oligomeric state appear to be an interesting consequence of the altered conformational landscape, but not its cause”. Additional HDX-MS experiments with the non-pooled monomeric and dimeric G320P species separately as well as the main SEC fraction of the G112P can help validating their claim.

In this new revision we have experimentally addressed all concerns of Reviewer #1.

More specifically:

In p.5 line 123-125 the authors state “Compared to wild type UraA, the fraction of dimers was reduced for UraA(G112P). Given that monomeric mutants of UraA are transport-incompetent, this may underlie the lack of transport activity observed for this mutant.”

I am not at all convinced with the quantification from one SEC profile of 5% and 3% of dimeric species for the WT and G112P proteins respectively in table S1. The authors are making conclusions based on a 2% difference without any replicates nor SDs...

How did the author integrate these humpy profiles? Due to the very low amount of dimeric species (if present) and the limitation in SEC sensitivity, the authors should include at least a quantification of triplicate SEC data with corresponding SD values in table S1. I presume they repeated the expression/purification a sufficient number of times to include this data.

The SEC profiles included in our original manuscript served to highlight the surprising and significant increase in the fraction of dimers for UraA(G320P) and UraA(P330G). We are happy to note that this is not a point of concern of Reviewer #1.

We did indeed note a reduction in the fraction of dimers compared to wild type for UraA(G112P), a mutant that played a peripheral role in our manuscript. Upon request of Reviewer #1, in this second revision we now include a more detailed analysis of this mutant.

We have added triplicate SEC data for each UraA variant tested in the absence [Fig. S4C] and presence of uracil [Fig. S4D] and we have quantified the monomeric and dimeric fractions [Fig. S4B, Table S1]. We have changed Fig. 1E to show a representative SEC profile for each mutant. All new and revised figures are shown below.

Based on this extensive analysis, we do not observe a significant difference in monomer/dimer ratio between UraA(WT) and UraA(G112P). We have thus corrected the respective paragraph (line 122-127) accordingly (see below).

Given the peripheral role of UraA(G112P) in our manuscript, this correction does not affect the main conclusions of our manuscript.

Decylmaltoside-solubilized wild type UraA eluted in two peaks previously assigned to UraA monomers and dimers¹³. For both wild type UraA and UraA(G112P), the monomeric species was the most abundant [Fig. 1E, Fig. S4B-C, Table S1]. In contrast, the dimer content was increased in UraA(G320P) and dimers were the dominant species in UraA(P330G). The presence of uracil did not alter the relative distribution of these populations [Fig. S4B, Fig. S4D]. Since constitutive UraA dimers have high transport activities¹³, it is not obvious why the UraA(G320P) and UraA(P330G) mutants are inactive.

Fig. 1: Functional analysis of inter-domain linker mutants in UraA. (A) Dimeric UraA structure (PDB:5XLS) with scaffold and transport domain in green or purple, respectively, and the interdomain-linkers in blue. (B) Topology plot of UraA (center) with color code as in (A). Upper and lower panels depict the sequence conservation of the extracellular and cytoplasmic interdomain-linkers in the UraA-subfamily, respectively, shown as sequence logo. (C) Absolute differences in dihedral angles between interdomain-linker residues in UraA_{IO} (PDB:3QE7) and UraA_{OCC} (PDB:5XLS). $\Delta\phi$ and $\Delta\psi$ are colored dark and light grey, respectively, and glycine and proline residues flanking the spacer helices are highlighted in red and orange, respectively. (D) *In vivo* uptake rates of [³H]-uracil by wild type UraA and interdomain-linker mutants upon expression in *E. coli* BW25113(Δ uraA) with technical replicates shown as scatter plot and derived mean values \pm SER as bars. A UraA variant with three alanine substitutions in the substrate binding site (E241A, H245A, and E290A) served as negative control. (E) Representative size-exclusion chromatograms of decylmaltoside-solubilized UraA variants in the absence of substrate. (F) Melting temperature of interdomain-linker mutants as determined by differential scanning fluorimetry in absence and presence of 1 mM uracil with technical replicates shown as scatter plot and derived mean values \pm SER as bars.

Fig. S4: (A) In-gel fluorescence (left panel) and coomassie staining (right panel) of the same SDS-PAGE gel loaded with the samples 1: control (UraA(WT) uninduced); 2: UraA(WT); 3: UraA(G112P); 4: UraA(P121G); 5: UraA(G320P) and 6: UraA(P330G) expressed from the pBXC3GH plasmid as C-terminal GFP fusion protein. (B) Dimer content of decylmaltoside-solubilized UraA variants in absence and presence of 1 mM uracil based on triplicate size-exclusion analysis as shown in panel C and D. The dimer content represents the ratio of the area under the dimer peak and the total area under the curve derived from Gaussian peak deconvolution. (C) Size-exclusion chromatograms of decylmaltoside-solubilized UraA variants in absence of uracil. (D) Size-exclusion chromatograms of decylmaltoside-solubilized UraA variants in presence of 1 mM uracil.

Table S1: Fractions of monomers and dimers observed for decylmaltoside-solubilized UraA variants by size-exclusion chromatography in the absence of uracil. Values represent averages and corresponding standard deviation of three technical replicates. The fraction of protomers migrating as monomers and dimers was calculated following peak deconvolution using Origin.

UraA variant	Monomer (%)	Dimer (%)
WT	79.2 ± 0.01	20.8 ± 0.01
G112P	80.2 ± 0.02	19.8 ± 0.02
G320P	55.1 ± 0.02	44.9 ± 0.02
P330G	17.6 ± 0.02	82.4 ± 0.02

In the same line of concerns, the authors state in their rebuttal: "Taken together, we conclude that the impact of the respective mutations lies in its effect on the conformational space of the transporter, and that the observed changes in the oligomeric state are a consequence of the altered conformational landscape, but not its cause." The oligomeric state of the G112P versus WT is not significantly impacted judging from the SEC data; therefore, I am not convinced that an altered conformational landscape will lead to a change of the oligomeric state. To consolidate their statement, the authors should analyze the conformation of the G112P mutant (that does not lead to a significant change in the oligomeric state) to check if the conformation is altered or not. Therefore, HDX-MS analysis of G112P will be a highly valuable experiment to consolidate the authors conclusions.

Our statement: "Taken together, we conclude that the impact of the respective mutations lies in its effect on the conformational space of the transporter, and that the observed changes in the oligomeric state are a consequence of the altered conformational landscape, but not its cause." was meant to address only the UraA(G320P) and UraA(P330G) mutants that we had characterized in detail. We now realize that it could be read as to include UraA(G112P), a mutant for which we did not study the conformational space and thus also did not reach any conclusion in this respect, as well. We regret this misunderstanding.

Nevertheless, as requested by Reviewer #1, we have now analyzed the conformational space of UraA(G112P) as well (shown in the modified supplementary figure S14; see below). For practical reasons, we have used the cysteine alkylation assay. These results suggest that UraA(G112P) is mostly in the inward-open conformation like wild type UraA. We have adjusted the respective paragraph (lines 207-212) accordingly (below).

Wild type UraA showed a two-fold higher degree of labeling with mPEG5K for the wide inward-open reporter compared to the outward-open reporter [Fig. 4B; Fig. S14]. Addition of the conformational probe Sy45 reduced labeling of the outward-open reporter due to the shifted conformational equilibrium towards the wide inward-open state. However, a concomitant increase in the degree of alkylation of the wide inward-open sensor was not observed. Similar observations were made for UraA(G112P) [Fig. S14], indicating that the conformational equilibrium of this mutant is not substantially altered.

While this additional data set on UraA(G112P) provides a more fine-grained view of the relevance of this region in the respective interdomain-linker, this result does not alter our conclusion that interdomain-linkers control conformational transitions in UraA.

Fig. S14: Degree of cysteine alkylation of L85C (black, control), L34C (red, outward-open reporter) and V248C (blue, wide inward-open reporter) in the presence and absence of Sy45 (grey). UraA variants were alkylated with mPEG5k over a time course of 1 hour and alkylation was quenched at indicated time points and quantified by mobility shift in SDS-PAGE and densitometry analysis. Data was normalized to the maximal alkylation obtained in the presence of 1% SDS (lane C). Shown are the mean of three technical replicates and standard error. For UraA(G320P) the monomer and dimer fractions from size-exclusion chromatography were analyzed separately and depicted as G320P_M and G320P_D. A two-tailed, unpaired t-test was performed to test for statistical significance (* $p < 0.05$, ** $p < 0.01$, *** $p < 0.001$).

In addition, SEC profile of G320P shows higher monomeric species compared to dimeric species; since the HDX-MS relate to the average of all states present in solution, the authors should repeat the experiments with the non-pooled monomeric G320P and dimeric G320P separately in order to demonstrate that the oligomeric state does not affect the overall HDX-MS data. Indeed, the differential HDX-MS profiles for apo versus uracil-bound proteins for WT and G320P mutant are very similar, and different compared to the P330G mutant (that gives mainly dimeric species); the WT VS. mutants comparison is also impacted by the fact that the analysis in the case of G320P was performed on the pooled monomeric and dimeric species.

As suggested by Reviewer #1, we have experimentally addressed the relation between the conformational space and oligomeric state by determining the conformational space of the non-pooled monomeric and dimeric fractions of UraA(G320P). For practical reasons, we have used the cysteine alkylation assay. These data have been added to supplementary figure S14 (above in full, below a cut-out).

For both the monomeric and dimeric species of UraA(G320P), we observe an overall higher accessibility for the outward-open reporter compared to the inward-open reporter, similar to the results obtained for the pooled UraA(G320P) monomers and dimers (Fig.4B). The accessibility of the outward-open sensor L34C shows no significant difference between monomers and dimers of UraA(G320P). The accessibility of the inward-open sensor V248C is significantly ($p < 0.01$) lower in the dimer, but the magnitude of this decrease is at most modest. In conclusion, the oligomeric state of UraA(G320P) does not affect the overall cysteine accessibility pattern.

Taken together, we expect that the outcome of these additional experiments have convincingly addressed the main major concern of Reviewer #1 and have consolidated our claim that “the observed changes in the oligomeric state appear to be an interesting consequence of the altered conformational landscape, but not its cause”. This result strengthens our conclusion that interdomain-linkers control conformational transitions in UraA.

We mention the analysis of the UraA(G320P) monomer and dimer fractions in lines 214-223 (below).

For UraA(G320P), we observed more alkylation of the outward-open sensor and significantly reduced accessibility of the wide inward-open sensor compared to wild type UraA [Fig 4B], suggesting an altered conformational equilibrium. Consistently, in the presence of Sy45 the population of UraA(G320P) occupying the wide inward-open state increased, indicating that the mutation did not arrest the protein in a specific conformation. Notably, the apparent change in conformational space occupancy was not caused by increased self-association of UraA(G320P) [Fig. 1E, Fig. S4B-C, Table S1], as separately isolated UraA(G320P) monomer and dimer fractions showed very similar cysteine accessibility profiles [Fig. S14]. The notion that UraA(G320P) populates a different section of the transporter’s conformational space is further supported by its four-fold higher uracil binding affinity [Fig. 4D]. Again, addition of Sy45 reverted the variant back to a state with an affinity very similar to wild type UraA¹³.

Other minor comments:

-The authors should clearly state in the text that they refer to the study from the Nieng Yan lab (Yu et al., 2017) to assume the oligomeric state based on the SEC data (p.5, line 121/122).

We added the requested changes to the paragraph (line 122-123):

*Decylmaltoside-solubilized wild type UraA eluted in two peaks that were previously assigned to UraA monomers and dimers*¹³.

- Fig. S16 is not really helpful to address the quality of the HDXMS data; instead, the authors should consider adding overall uptake plots of the different peptides in compared states, which provides a better and complete representation.

We thank the reviewer for pointing this out. We added overall uptake plots of three different peptides for the UraA(WT), UraA(G320P) and UraA(P330G) datasets in the new supplementary figure 18 (Fig. S18).

Fig. S18: Relative deuterium uptake of 3 selected peptides for each UraA variant in presence and absence of uracil at four different time points (0.5 min, 6 min, 15 min and 45 min). Datapoints represent mean values \pm SER from four technical replicates.

Reviewer #2 (Remarks to the Author):

I have looked through the rebuttal and reread the paper. The authors have answered the major concerns. The only further comment that I have regards the presence of uracil in the crystal structure. This does not really influence any of the major conclusions of the paper, but from the new supplementary Figure 8, it is not obvious that there is any difference between the "uracil bound" and the density of the Tris. There is 50mM Tris in both the crystal conditions. Is it possible that Tris is present in both cases? Density should really be shown from maps calculated before the addition of the ligands. If there is any uncertainty in interpretation this should be acknowledged in the text.

We are pleased to read that we have satisfactorily addressed the major concerns of Reviewer #2.

We have also addressed the minor remark and per suggestion of the reviewer, we now use Fo-Fc omit maps calculated in absence of the ligands for supplementary figure 8 (revised figure below). We agree with the reviewer that at the given resolution uracil and tris are indistinguishable. We therefore performed additional experiments to support the identity of the modeled ligands in our structures.

Using the scintillation proximity assay, we have quantified uracil binding in the presence of 50 mM Tris, which equals the Tris concentration in the crystallization condition. Under these conditions, we observe a two-fold decrease in uracil affinity ($K_{D,app} = 379 \pm 100$ nM instead of $K_D = 148 \pm 36$ nM).

Based on the identical position of the electron density of the ligand in the two structures, we assume Tris to be a competitive inhibitor. Thus, the K_I of Tris can be calculated as follows: $K_{D,app} = K_D(1 + I/K_I)$, resulting in a K_I of 32 ± 20 mM (see detailed breakdown in panel E below).

Based on the uracil affinity of UraA(G320P-Sy45) ($K_D = 144$ nM), the Tris binding affinity ($K_I = 32$ mM) and the concentrations in the crystallization condition (uracil: 1 mM, Tris: 50 mM) more than 99% of the UraA molecules are expected to be in complex with uracil.

We therefore conclude that it is unlikely that Tris is bound in the +1 mM uracil condition.

However, in absence of uracil, the concentration of Tris in the crystallization condition is sufficiently high to allow Tris binding to the substrate binding site of UraA. We have added this analysis to the supplementary figure 8 [Fig. S8].

D

dataset	UraA(G320P)-Sy45	UraA(G320P)-Sy45 Uracil
Uracil ₀ (mM)	-	1
K _D (nM)	144 ± 18	144 ± 18
Tris ₀ (mM)	50	50
K _I (mM)	32 ± 20	32 ± 20
Uracil bound (%)	-	99.9 ± 1.8
Tris bound (%)	60.9 ± 14.8	0.022 ± 0.003

E

$$(1) \quad K_{D,app} = K_D(1 + \text{Tris}/K_I)$$

$$(2) \quad \frac{\text{UraA}_{\text{Uracil}}}{\text{UraA}_0} = \frac{(\text{Uracil}_0 + K_D(1 + \text{Tris}_0/K_I) + \text{UraA}_0) - \sqrt{(\text{Uracil}_0 + K_D(1 + \text{Tris}_0/K_I) + \text{UraA}_0)^2 - 4\text{Uracil}_0\text{UraA}_0}}{2\text{UraA}_0}$$

$$(3) \quad \frac{\text{UraA}_{\text{Tris}}}{\text{UraA}_0} = \frac{(\text{Tris}_0 + K_I + \text{UraA}_0) - \sqrt{(\text{Tris}_0 + K_I + \text{UraA}_0)^2 - 4\text{Tris}_0\text{UraA}_0}}{2\text{UraA}_0}$$

Fig. S8: Substrate binding site of UraA(G320P)-Sy45 apo (**A**) and uracil (**B**) structures. The Fo-Fc omit maps are shown as blue mesh and contoured at 3 σ . Though the UraA(G320P)-Sy45 complex was co-crystallized in absence of uracil, additional electron density in the substrate binding site was observed. As substantial co-purification of uracil could be excluded based on the observed thermostabilization upon addition of uracil [**Fig. 1E**], we modeled the buffer compound Tris into the density. (**C**) Scintillation proximity assay of UraA wild type in absence and presence of 50 mM Tris as indicated with three technical replicates shown as grey scatter and derived SER shown as black error bars. Scintillation data was fitted in Origin with a binding curve for homologous competition to calculate the dissociation constants. (**D**) Uracil- and Tris-bound fractions of UraA(G320P) present under the crystallization conditions as calculated with equations shown in (**E**). Based on the identical localization of the ligand electron density in the two structures we assumed Tris to be a competitive inhibitor. The K_I of Tris was calculated using equation (E1). Based on the affinity for uracil (K_D = 144 nM) and Tris (K_I = 32 mM) and the concentrations in the crystallization condition (uracil: 1 mM, Tris: 50 mM, UraA: 0.16 mM) the uracil-bound fraction was calculated with equation (E2) and exceeded 99%. The Tris bound fraction in absence of uracil was calculated to be 61 ± 15% with equation (E3).

REVIEWERS' COMMENTS

Reviewer #1 (Remarks to the Author):

The authors addressed my main concerns in the updated version of the manuscript.

Reviewer #2 (Remarks to the Author):

The authors have comprehensively addressed the concerns of the reviewers.